# How Far Is Video Generation from World Model: A Physical Law Perspective

## Abstract

OpenAI's Sora highlights the potential of video generation for developing world models that adhere to fundamental physical laws. However, the ability of video generation models to discover such laws purely from visual data without human priors can be questioned. A world model learning the true law should give predictions robust to nuances and correctly extrapolate on unseen scenarios. In this work, we evaluate across three key scenarios: in-distribution, out-of-distribution, and combinatorial generalization. We developed a 2D simulation testbed for object movement and collisions to generate videos deterministically governed by one or more classical mechanics laws. This provides an unlimited supply of data for large-scale experimentation and enables quantitative evaluation of whether the generated videos adhere to physical laws. We trained diffusion-based video generation models to predict object movements based on initial frames. Our scaling experiments show perfect generalization within the distribution, measurable scaling behavior for combinatorial generalization, but failure in out-of-distribution scenarios. Further experiments reveal two key insights about the generalization mechanisms of these models: (1) the models fail to abstract general physical rules and instead exhibit "case-based" generalization behavior, *i.e.*, mimicking the closest training example; (2) when generalizing to new cases, models are observed to prioritize different factors when referencing training data: color > size > velocity > shape. Our study suggests that scaling alone is insufficient for video generation models to uncover fundamental physical laws, despite its role in Sora's broader success.

## 1 Introduction

Foundation models (Bommasani et al., 2021) have emerged remarkable capabilities by scaling the model and data to an unprecedented scale (Brown, 2020; Kaplan et al., 2020). As an example, OpenAI's Sora (Brooks et al., 2024) not only generates high-fidelity and surreal videos, but also has sparked a new surge of interest in studying world models (Yang et al., 2023).

> "*Scaling video generation models is a promising path towards building general purpose simulators of the physical world.*" — Sora Report (Brooks et al., 2024)

World simulators are receiving broad attention from robotics (Yang et al., 2023) and autonomous driving (Hu et al., 2023) for the ability to generate realistic data and accurate simulations. These models are required to comprehend fundamental physical laws to produce data that extends beyond the training corpus and to guarantee precise simulation. However, it remains an open question whether video generation can discover such rules merely by observing videos, as Sora does. We aim to provide a systematic study to understand the critical role and limitation of scaling in physical law discovery.

It is challenging to determine whether a video model has learned a law instead of merely memorizing the data. Since the model's internal knowledge is inaccessible, we can only infer the model's understanding by examining its predictions on unseen scenarios, *i.e.*, its generalization ability. We propose a categorization (Figure 1) for comprehensive evaluation based on the relationship between training and testing data in this paper. *In-distribution* (ID) generalization assumes that training and testing data are independent and identically distributed (*i.i.d.*). *Out-of-distribution* (OOD) generalization, on the other hand, refers to the model's performance on testing data that come from a different distribution than the training data, particularly when latent parameters fall outside the range seen during training. Human-level physical reasoning can easily extrapolate OOD and predict physical processes without encountering the exact same scenario before. Additionally, we also examine a special OOD capacity called *combinatorial* generalization, which assesses whether

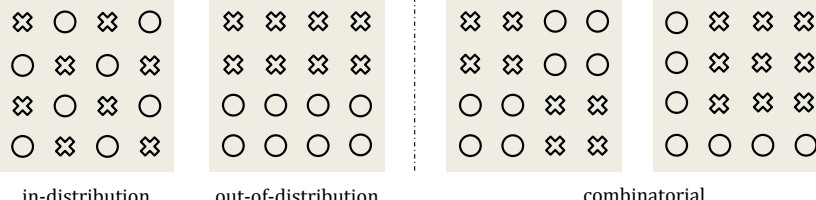

in-distribution      out-of-distribution         combinatorial

Figure 1: Categorization of generalization patterns.◯ denotes training data. ✕ denotes testing data.

a model can combine two distinct concepts in a novel way, a trait often considered essential for foundation models in advancing toward artificial general intelligence (AGI) (Du & Kaelbling, 2024).

Moreover, real-world videos typically contain complex, non-rigid objects and motions, which present significant challenges for quantitative evaluation and even human validation. The rich textures and appearances in such videos can act as confounding factors, distracting the model from focusing on the underlying physics. To mitigate these issues, we specifically focus on classical mechanics and develop a 2D simulator with objects represented by simple geometric shapes. Each video depicts the motion or collision of these 2D objects, governed entirely by one or two fundamental physical laws, given the initial frames. This simulator allows us to generate large-scale datasets to support the scaling of video generation models. Additionally, we have developed a tool to infer internal states (*e.g.*, the position and size) of each object in the generated video from pixels. This enables us to establish quantitative evaluation metrics for physical law discovery.

We begin by investigating how scaling video generation models affects ID and OOD generalization. We select three fundamental physical laws for simulation: *uniform linear motion* of a ball, *perfectly elastic collision* between two balls, and *parabolic motion* of a ball. We scale the dataset from 30K to 3 million examples and increase the video diffusion model's parameters from 22M to 310M. Consistently, we observe that the model achieves near-perfect ID generalization across all tasks. However, the OOD generalization error does not improve with increased data and model size, revealing the limitations of scaling video generation models in handling OOD data. For combinatorial generalization, we design an environment that involves multiple objects undergoing free fall and collisions to study their interactions. Every time, four objects from eight are selected to create a video. In total, 70 combinations ($C_8^4$) are possible. We use 60 of them for training and 10 for testing. We train models by varying the number of training data from 600K to 6M. We manually evaluate the generated test samples by labeling them as "abnormal" if the video looks physically implausible. The results demonstrate that scaling the data substantially reduces the percentage of abnormal cases, from 67% to 10%. This suggests that scaling is critical for improving combinatorial generalization.

Our empirical analysis reveals two intriguing properties of the generalization mechanism in video generation models. First, these models can be easily biased by "deceptive" examples from the training set, leading them to generalize in a "case-based" manner under certain conditions. This phenomenon, also observed in large language models (Hu et al., 2024), describes a model's tendency to reference similar training cases when solving new tasks. For instance, consider a video model trained on data of a high-speed ball moving in uniform linear motion. If data augmentation is performed by horizontally flipping the videos, thereby introducing reverse-direction motion, the model may generate a scenario where a low-speed ball reverses direction after the initial frames, even though this behavior is not physically correct. Second, we explore how different data attributes compete during the generalization process. For example, if the training data for uniform motion consists of red balls and blue squares, the model may transform a red square into a ball immediately after the conditioning frames. This behavior suggests that the model prioritizes color over shape. Our pairwise analysis reveals the following prioritization hierarchy: color > size > velocity > shape. This ranking could explain why current video generation models often struggle with maintaining object consistency. We hope these findings provide valuable insights for future research in video generation and world models.

## 2 DISCOVERING PHYSICS LAWS WITH VIDEO GENERATION

### 2.1 PROBLEM DEFINITION

In this section, we aim to establish the framework and define the concept of physical laws discovery in the context of video generation. In classical physics, laws are articulated through mathematical equations that predict future state and dynamics from initial conditions. In the realm of video-based

observations, each frame represents a moment in time, and the prediction of physical laws corresponds to generating future frames conditioned on past states.

Consider a physical procedure which involves several latent variables $\boldsymbol{z} = (z_1, z_2, \ldots, z_k) \in \mathcal{Z} \subseteq \mathbb{R}^k$, each standing for a certain physical parameter such as velocity or position. By classical mechanics, these latent variables will evolve by differential equation $\dot{\boldsymbol{z}} = F(\boldsymbol{z})$. In discrete version, if time gap between two consecutive frames is $\delta$, then we have $\boldsymbol{z}_{t+1} \approx \boldsymbol{z}_t + \delta F(\boldsymbol{z}_t)$. Denote rendering function as $R(\cdot) : \mathcal{Z} \mapsto \mathbb{R}^{3 \times H \times W}$ which render the state of the world into an image of shape $H \times W$ with RGB channels. Consider a video $V = \{I_1, I_2, \ldots, I_L\}$ consisting of $L$ frames that follows the classical mechanics dynamics. The physical coherence requires that there exists a series of latent variables which satisfy following requirement: 1) $\boldsymbol{z}_{t+1} = \boldsymbol{z}_t + \delta F(\boldsymbol{z}_t), t = 1, \ldots, L-1$. 2) $I_t = R(\boldsymbol{z}_t), \quad t = 1, \ldots, L$. We train a video generation model $p$ parametried by $\theta$, where $p_\theta(I_1, I_2, \ldots, I_L)$ characterizes its understanding of video frames. We can predict the subsequent frames by sampling from $p_\theta(I'_{c+1}, \ldots I'_L \mid I_1, \ldots, I_c)$ based on initial frames' condition. The variable $c$ usually takes the value of 1 or 3 depends on tasks. Therefore, physical-coherence loss can be simply defined as $-\log p_\theta(I_{c+1}, \ldots, I_L \mid I_1, \ldots, I_c)$. It measures how likely the predicted value will cater to the real world development. The model must understand the underlying physical process to accurately forecast subsequent frames, which we can quantatively evaluate whether video generation model correctly discover and simulate the physical laws.

## 2.2 Video Generation Model

In this paper, we focus exclusively on video generation models, leaving discussions on other modeling methods for future work. Following Sora (Brooks et al., 2024), we adopt the Variational Auto-Encoder (VAE) and DiT architectures for video generation. The VAE compresses videos into latent representations both spatially and temporally, while the DiT models the denoising process. This approach shows strong scalability and achieves promising results in generating high-quality videos.

**VAE Model.** We employ a (2+1)D-VAE to project videos into a latent space. Starting with the SD1.5-VAE structure, we extend it into a spatiotemporal autoencoder using 3D blocks (Yu et al., 2023b). All parameters of the (2+1)D-VAE are pretrained on high-quality image and video data to maintain strong appearance modeling while enabling motion modeling. More details are provided in Appendix A.3.1. In this paper, we *fix* the pretrained VAE encoder and use it as a video compressor. Results in Appendix A.3.2 confirm the VAE's ability to accurately encode and decode the physical event videos. This allows us to focus solely on training the diffusion model to learn the physical laws.

**Diffusion model.** Given the compressed latent representation from the VAE model, we flatten it into a sequence of spacetime patches, as transformer tokens. Notably, self-attention is applied to the entire spatio-temporal sequence of video tokens, without distinguishing between spatial and temporal dimensions. For positional embedding, a 3D variant of RoPE (Su et al., 2024) is adopted. As stated in Sec.2.1, our video model is conditioned on the first $c$ frames. The $c$-frame video is zero-padded to the same length as the full physical video. We also introduce a binary mask "video" by setting the value of the first $c$ frames to 1, indicating those frames are the condition inputs. The noise, condition and mask videos are concatenated along the channel dimension to form the final input to the model.

## 2.3 On the Verification of Learned Laws

Suppose we have a video generation model learned based on the above formulation. How do we determine if the underlying physical law has been discovered? A well-established law describes the behavior of the natural world, *e.g.*, how objects move and interact. Therefore, a video model incorporating true physical laws should be able to withstand experimental verification, producing reasonable predictions under any circumstances, which demonstrates the model's generalization ability. To comprehensively evaluate this, we consider the following categorization of generalization (see Figure 1) within the scope of this paper: 1) **In-distribution** (ID) generalization describes the setting where training data and testing data are from the same distribution. In our case, both training and testing data follow the same law and are located in the same domain. 2) A human who has learned a physical law can easily extrapolate to scenarios that have never been observed before. This ability is referred to as **out-of-distribution** (OOD) generalization. Although it sounds challenging, this evaluation is necessary as it indicates whether a model can learn principled rules from data. 3) Moreover, there is a situation between ID and OOD, which has more practical value. We call

this **combinatorial** generalization, representing scenarios where every "concept" or object has been observed during training, but not their every combination. It examines a model's ability to effectively combine relevant information from past experiences in novel ways. A similar concept has been explored in LLMs (Riveland & Pouget, 2024), which demonstrated that models can excel at linguistic instructing tasks by recombining previously learned components, without task-specific experience.

## 3 IN-DISTRIBUTION AND OUT-OF-DISTRIBUTION GENERALIZATION

In this section, we study the correlation between ID and OOD generalization and model or data scaling, focusing on video generation models. We use deterministic tasks governed by basic kinematic equations, as they allow clear definitions of ID/OOD and straightforward quantitative error evaluation.

### 3.1 FUNDAMENTAL PHYSICAL SCENARIOS

Specifically, we consider three physical scenarios illustrated in Fig. 2. 1) **Uniform Linear Motion:** A colored ball moves horizontally with a constant velocity. This is used to illustrate *the law of Intertia*. 2) **Perfectly Elastic Collision:** Two balls with different sizes and speeds move horizontally toward each other and collide. The underlying physical law is *the conservation of energy and momentum*. 3) **Parabolic Motion:** A ball with a initial horizontal velocity falls due to gravity. This represents *Newton's second law of motion*. Each motion is determined by its initial frames.

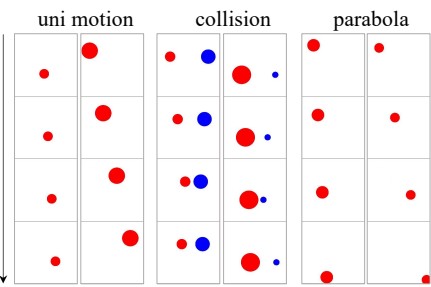

uni motion    collision    parabola

**Training data generation.** We use Box2D to simulate kinematic states for various scenarios and render them as videos, with each scenario having 2-4 degrees of freedom (DoF), such as the balls' initial velocity and mass. An in-distribution range is defined for each DoF. We generate training datasets of 30K, 300K, and 3M videos by uniformly sampling a high-dimensional grid within these ranges. All balls have the same density, so their mass is inferred from their size. Gravitational acceleration is constant in parabolic motion for consistency. Initial ball positions are randomly initialized within the visible range. Further details are provided in Appendix A.4.1.

Figure 2: Downsampled video visualization. The arrow indicates the progression of time.

Table 1: Details of DiT model sizes.

**Test data generation.** We evaluate the trained model using both ID and OOD data. For ID evaluation, we sample from the same grid used during training, ensuring that no specific data point is part of the training set. OOD evaluation videos are generated with initial radius and velocity values outside the training range. There are various types of OOD setting, *e.g.* velocity/radius-only or both OOD. Details are provided in Appendix A.4.1.

| Model | Layers | Hidden size | Heads | #Param |
|---|---|---|---|---|
| DiT-S | 12 | 384 | 6 | 22.5M |
| DiT-B | 12 | 768 | 12 | 89.5M |
| DiT-L | 24 | 1024 | 16 | 310.0M |
| DiT-XL | 28 | 1152 | 16 | 456.0M |

**Models.** For each scenario, we train models of varying sizes from scratch, as shown in Table 1. This ensures that the outcomes are not influenced by uncontrollable pretrain data. The first three frames are provided as conditioning, which is sufficient to infer the velocity of the balls and predict the subsequent frames. Diffusion model is trained for 100K steps using 32 Nvidia A100 GPUs with a batch size of 256, which was sufficient for convergence, as a model trained for 300K steps achieves a similar performance. We keep the pretrained VAE fixed. Each video consists of 32 frames with a resolution of 128x128. We also experimented with a 256x256 resolution, which yielded a similar generalization error but significantly slowed down the training process.

**Evaluation metrics.** We observed that the learned models are able to generate balls with consistent shapes. To obtain the center positions of the $i$-th ball in the generated videos, $x_t^i$, we use a heuristic algorithm based on the mean of colored pixels, distinguishing the balls by color. To ensure the correctness of $x_t^i$, we exclude frames with part of a ball out of view, yielding valid frames $T$. For collision scenarios, only the frames after the collision are considered. We then compute the velocity of each ball, $v_t^i$, at each moment by differentiating their positions. The *error* for a video is defined as: $e = \frac{1}{N|T|} \sum_{i=1}^{N} \sum_{t \in T} \left| v_t^i - \hat{v}_t^i \right|$, where $v_t^i$ is the computed velocity at time $t$, $\hat{v}_t^i$ is the ground-truth velocity in the simulator, $N$ is the number of balls, and $|T|$ is the number of valid frames.

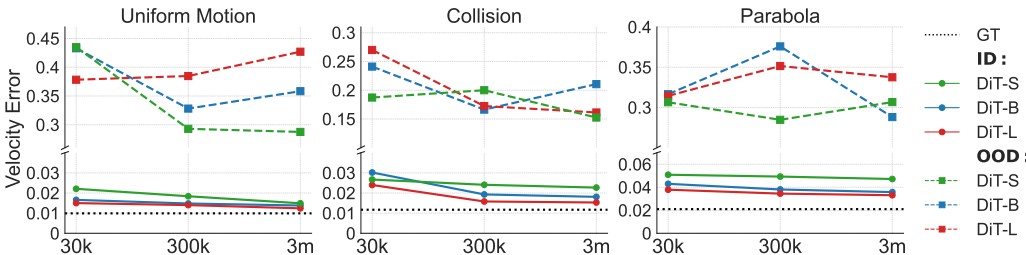

Figure 3: The error in the velocity of balls between the ground truth state in the simulator and the values parsed from the generated video by the diffusion model, given the first 3 frames.

**Baseline.** We calculate the error between the ground truth velocity and the parsed values from the ground truth video, referred to as *Groundtruth* (GT). This represents the system error—caused by parsing video into velocity—and defines the minimum error a model can achieve.

### 3.2 PERFECT ID AND FAILED OOD GENERALIZATION

For **in-distribution** (ID) generalization in Figure 3, increasing the model size (DiT-S to DiT-L) or the data amount (30K to 3M) consistently decreases the *velocity error* across all three tasks, strongly evidencing the importance of scaling for ID generalization. Take the uniform motion task as an example: the DiT-S model has a velocity error of 0.022 with 30K data, while DiT-L achieves an error of 0.012 with 3M data, very close to the error of 0.010 obtained with ground truth video.

However, the results differ significantly for **out-of-distribution** (OOD) predictions. First, OOD velocity errors are an order of magnitude higher than ID errors in all settings. For example, the OOD error for the DiT-L model on uniform motion with 3M data is 0.427, while the ID error is just 0.012. Second, scaling up the training data and model size has little or negative impact on reducing this prediction error. The variation in velocity error is higly random as data or model size changes, *e.g.*, the error for DiT-B on uniform motion is 0.433, 0.328 and 0.358, with data amounts of 30K, 300K and 3M. We also trained DiT-XL on the uniform motion 3M dataset but observed no improvement in OOD generalization. As a result, we did not pursue training of DiT-XL on other scenarios or datasets constrained by resources. These findings suggest the inability of scaling to perform reasoning in OOD scenarios. The sharp difference between ID and OOD settings further motivates us to study the generalization mechanism of video generation in Section 5.2.

## 4 COMBINATORIAL GENERALIZATION

In Section 3, video generation models failed to reason in OOD scenarios. This is understand-able—deriving precise physical laws from data is difficult for both humans and models. For example, it took scientists centuries to formulate Newton's three laws of motion. However, even a child can in-tuitively predict outcomes in everyday situations by combining elements from past experiences. This ability to combine known information to predict new scenarios is called *combinatorial generalization*. In this section, we evaluate the combinatorial abilities of diffusion-based video models.

### 4.1 COMBINATORIAL PHYSICAL SCENARIOS

We selected the PHYRE simulator (Bakhtin et al., 2019) as our testbed—a 2D environment involves multiple objects to free fall then collide with each other, forming complex physical interactions. It features diverse object types, including balls, jars, bars, and walls, which can be either fixed or dynamic. This enables complex interactions such as collisions, parabolic trajectories, rotations, and friction to occur simultaneously within a video. Despite this complexity, the underlying physical laws are deterministic, allowing the model to learn the laws and predict unseen scenarios.

**Training Data.** There are eight types of objects considered, including two dynamic gray balls, a group of fixed black balls, a fixed black bar, a dynamic bar, a group of dynamic standing bars, a dynamic jar, and a dynamic standing stick. Each task contains one red ball and four randomly chonsen objects from the eight types, resulting in $C_8^4 = 70$ unique templates. See Figure 4 for examples.

Each template was initialized with random sizes and positions for four objects, generating 100K videos to cover a range of possible scenarios. To explore the model's combinatorial ability and scaling effects, we structured the training data at three levels: a minimal set of 6 templates (0.6M videos) that includes all types of two-object interactions among the eight object types, and larger sets with 30 and 60 templates (3M/6M videos), with the 60-template set nearly covering the entire template space. The minimal training set places the highest demand on the model's ability for compositional generalization.

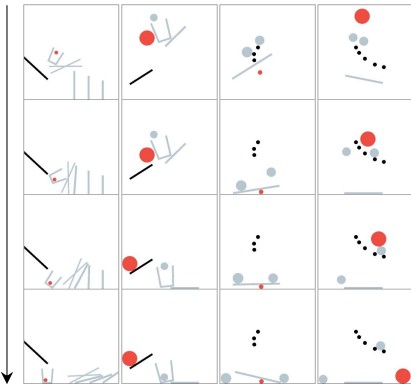

Figure 4: Downsampled videos. The black objects are fixed and others are dynamic.

**Test Data.** For each training template, we reserve a small set of videos to create the *in-template* evaluation set. Additionally, 10 unused templates are reserved for the *out-of-template* evaluation set to assess the model's ability to generalize to new combinations not seen during training.

**Models.** The first frame is used as the conditioning for video generation since the inital objects are static. We found that smaller models like DiT-S struggled with complex videos, so we primarily used DiT-B and DiT-XL. All models were trained for long 1000K gradient steps on 64 Nvidia A100 GPUs with a batch size of 256, ensuring near convergence. To better capture the complexity of physical events, we increased the resolution to 256x256 with 32 frames.

**Evaluation Metrics.** We use several metrics to assess the fidelity of generated videos compared to the ground truth. Frechet Video Distance (FVD) (Unterthiner et al., 2018) calculates feature distances between generated and real videos using features from Inflated-3D ConvNets (I3D) pretrained on Kinetics-400 (Carreira & Zisserman, 2017). SSIM and PSNR (Wang et al., 2004) are pixel-level metrics: SSIM evaluates brightness, contrast, and structural similarity, while PSNR measures the ratio between peak signal and mean squared error, both averaged across frames. LPIPS (Zhang et al., 2018) gauges perceptual similarity between image patches. We include human evaluations, reporting the *abnormal ratio* of generated videos that violate physical laws assessed by humans.

## 4.2 SCALING LAW OBSERVED FOR COMBINATORIAL GENERALIZATION

Table 2: Combinatorial generalization results. The results are presented in the format of {*in-template result*} / {*out-of-template result*}.

| Model | #Templates | FVD (↓) | SSIM (↑) | PSNR (↑) | LPIPS (↓) | Abnormal (↓) |
|---|---|---|---|---|---|---|
| DiT-XL | 6 | 18.2 / 22.1 | **0.973** / 0.943 | **32.8** / 25.5 | **0.028** / 0.082 | 3% / 67% |
| DiT-XL | 30 | 19.5 / 19.7 | 0.973 / 0.950 | 32.7 / 27.1 | 0.028 / 0.065 | 3% / 18% |
| DiT-XL | 60 | **17.6** / **18.7** | 0.972 / **0.951** | 32.4 / **27.3** | 0.030 / **0.062** | **2% / 10%** |
| DiT-B | 60 | 18.4 / 21.4 | 0.967 / 0.949 | 30.9 / 27.0 | 0.035 / 0.066 | 3% / 24% |

It requires higher resolution, much more training iterations, and larger model sizes to perform well on this task due to increased complexity. Consequently, we are unable to conduct a comprehensive sweep of all data and model size combinations as in Section 3. Therefore, we start with the largest model, DiT-XL, to study data scaling behavior for combinatorial generalization. As shown in Table 2, when the number of templates increases from 6 to 60, all metrics improve on the out-of-template testing sets. Notably, the abnormal rate for human evaluation significantly reduces from 67% to 10%. Conversely, the model trained with 6 templates achieves the best SSIM, PSNR, and LPIPS scores on the in-template testing set. This can be explained by the fact that each training example in the 6-template set is exposed ten times more frequently than those in the 60-template set, allowing it to better fit the in-template tasks associated with template 6. Furthermore, we conducted an additional experiment using a DiT-B model on the full 60 templates to verify the importance of model scaling. As expected, the abnormal rate increases to 24%. These results suggest that both model capacity and coverage of the combination space are crucial for combinatorial generalization. This insight implies that scaling laws for video generation should focus on increasing combination diversity, rather than merely scaling up data volume. Some generated videos can be found in Figure 21 and Figure 22.

## 5 DEEPER ANALYSIS

In this section, we aim to investigate the generalization mechanism of a video generation model, through systemic experimental designs. Based on the findings, we try to identify certain patterns in combinatorial generalization that might be helpful in harnessing or prompting the models.

### 5.1 UNDERSTANDING GENERALIZATION FROM INTERPOLATION AND EXTRAPOLATION

The generalization ability of a model roots from its interpolation and extrapolation capability (Xu et al., 2020; Balestriero et al., 2021). In this section, we design experiments to explore the limits of these abilities for a video generation model. We design datasets which deliberately leave out some latent values, i.e. velocity. After training, we test model's prediction on both seen and unseen scenarios. We mainly focus on uniform motion and collision processes.

**Uniform Motion.** We create a series of training sets, where a certain range of velocity is absent. Each set contains 200K videos to ensure fairness. As shown in Figure 5 (1)-(2), with a large gap in the training set, the model tends to generate videos where the velocity is either high or low to resemble training data when initial frames show middle-range velocities. We find that the video generation model's OOD accuracy is closely related to the size of the gap, as seen in Figure 5 (3), when the gap is reduced, the model correctly interpolates for most of OOD data. Moreover, as shown in Figure 5 (4) and (5), when a subset of the missing range is reintroduced (without increasing data amount), the model exhibits stronger interpolation abilities.

**Collision.** It involves multiple variables, and is more challenging since the model has to learn a two-dimensional non-linear function. Specifically, we exclude one or more square regions from the training set of initial velocities for two balls and then assess the velocity prediction error after the collision. For each velocity point, we sample a grid of radius parameters to generate multiple video cases and compute the average error. As shown in Figure 6 (1)-(2), an interesting phenomenon occurs. The video generation model's extrapolation error demonstate an intriguing discrepancy among OOD points: For the OOD velocity combinations that lie within the convex hull of the training set, *i.e.*, the internal red squares in the yellow region, the model generalizes well. However, the model experiences large errors when the latent values lie in the exterior space of the training set's convex hull.

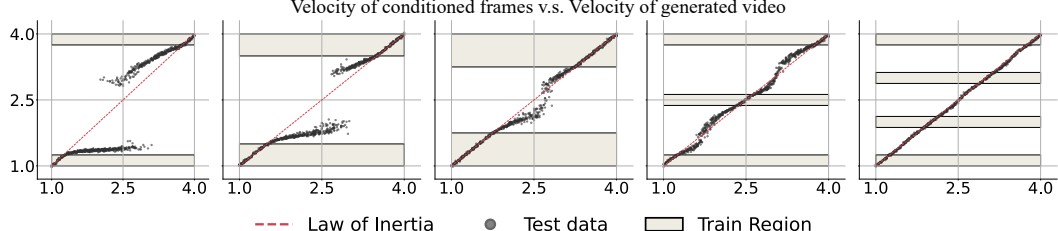

Velocity of conditioned frames v.s. Velocity of generated video

- - - Law of Inertia    ● Test data    ▭ Train Region

Figure 5: Uniform motion video generation. Models are trained on datasets with a missing middle velocity range. For example, in the first figure, training velocities cover $[1.0, 1.25]$ and $[3.75, 4.0]$, excluding the middle range. When evaluated with velocity condition from the missing range $[1.25, 3.75]$, the generated velocity tends to shift away from the initial condition, breaking the Law of Inertia.

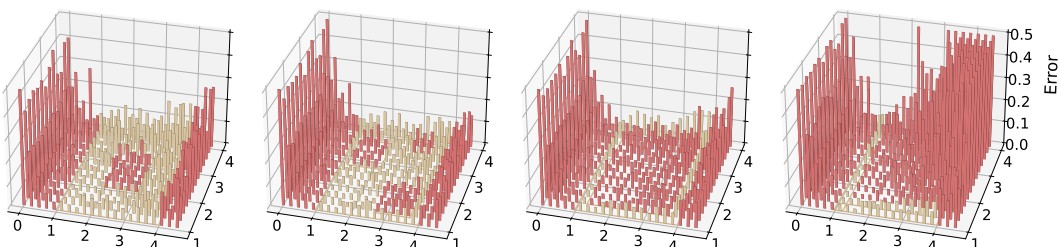

Figure 6: Collision video generation. Models are trained on the yellow region and evaluated on data points in both the yellow (ID) and red (OOD) regions. When the OOD range is surrounded by the training region, the OOD generalization error remains relatively small and comparable to the ID error.

## 5.2 MEMORIZATION OR GENERALIZATION

Previous work (Hu et al., 2024) indicates that LLMs rely on memorization, reproducing training cases during inference instead of learning the underlying rules for tasks like addition arithmetic. In this section, we investigate whether video generation models display similar behavior, memorizing data rather than understanding physical laws, which limits their generalization to unseen data.

We train our model on uniform motion videos with velocities $v \in [2.5, 4.0]$, using the first three frames as input conditions. Two training sets are used: *Set-1* only contains balls moving from left to right, while *Set-2* includes movement in both direction, by using horizontal flipping at training time. At evaluation, we focus on low-speed balls ($v \in [1.0, 2.5]$), which were not present in the training data. As shown in Figure 7, the *Set-1* model generates videos with only positive velocities, biased toward the high-speed range. In contrast, the *Set-2* model occasionally produces videos with negative velocities, as highlighted by the green circle. For instance, a low-speed ball moving from left to right may suddenly reverse direction after its condition frames. This could occur since the model identifies reversed training videos as the closest match for low-speed balls. This distinction between the two models suggests that the video generation model is influenced by "deceptive" examples in the training data. Rather than abstracting universal rules, the model appears to rely on memorization, and case-based imitation for OOD generalization.

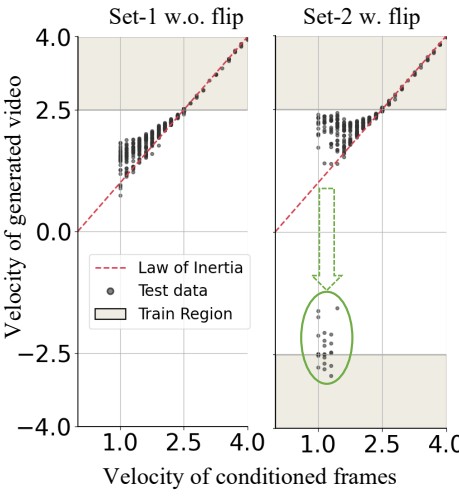

Figure 7: The example of uniform motion illustrating memorization.

## 5.3 HOW DOES DIFFUSION MODEL RETRIEVE DATA?

We aim to investigate the ways a video model performs *case matching*—identifying close training examples for a given test input. We use *uniform linear motion* for this study. Specifically, we compare four attributes, *i.e.*, color, shape, size, and velocity, in a pairwise manner. Through comparisons, we seek to determine the model's preference for relying on specific attributes in case matching.

Every attribute has two disjoint sets of values. For each pair of attributes, there are four types of combinations. We use two combinations for training and the remaining two for testing. For example, we compare color and shape in Figure 8 (1). Videos of red balls and blue squares with the same range of size and velocity are used for training. At test time, a blue ball changes shape into a square immediately after the condition frames, while a red square transforms into a ball. We observed no exceptions on 1,400 test cases, showing that the model prioritizes color over shape for case matching. A similar trend is observed in the comparisons of size vs. shape and velocity vs. shape, as illustrated in Figure 8 (2)-(3), indicating that shape is the least prioritized attribute. This suggests that diffusion-based video models inherently favor other attributes over shape, which may explain why current open-set video generation models usually struggle with shape preservation.

The other three attribute pairs are presented in Figure 9. For the continuous attributes of velocity and size, we selected two distinct values at the extreme boundaries to create binary comparisons: 1.0 and 4.0 for velocity, and 0.7 and 1.4 for size. For the comparison of color *vs.* size, the training set consists of: (1) red, small balls and (2) blue, large balls. During evaluation, across 202 evaluation cases, size consistently transformed while color remained fixed, clearly demonstrating that color > size. Similar observations are evident in Figure 9 (2) and (3), leading to the conclusion that color > velocity and size > velocity. Based on the above analysis, we establish the prioritization order as color > size > velocity > shape.

## 5.4 HOW DOES COMPLEX COMBINATORIAL GENERALIZATION HAPPEN?

In Section 4, we show that scaling the data coverage can boost combinatorial generalization. But what kind of data can actually enable conceptually-combinable video generation? In this section, we idenfify three foundamental combinatorial patterns through experimental design.

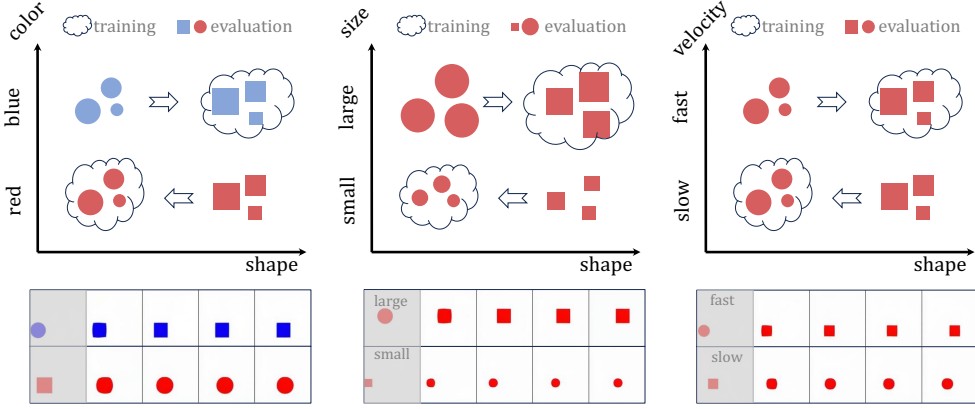

Figure 8: Uniform motion. (1) Color *v.s.* shape, (2) Size *v.s.* shape, (3) Velocity *v.s.* shape. The arrow ⇒ signifies that the generated videos shift from their specified conditions to resemble similar training cases. For example, in the first figure, the model is trained on videos of blue balls and red squares. When conditioned with a blue ball, as shown in the bottom, it transforms into a blue square, i.e., mimicking the training case by color.

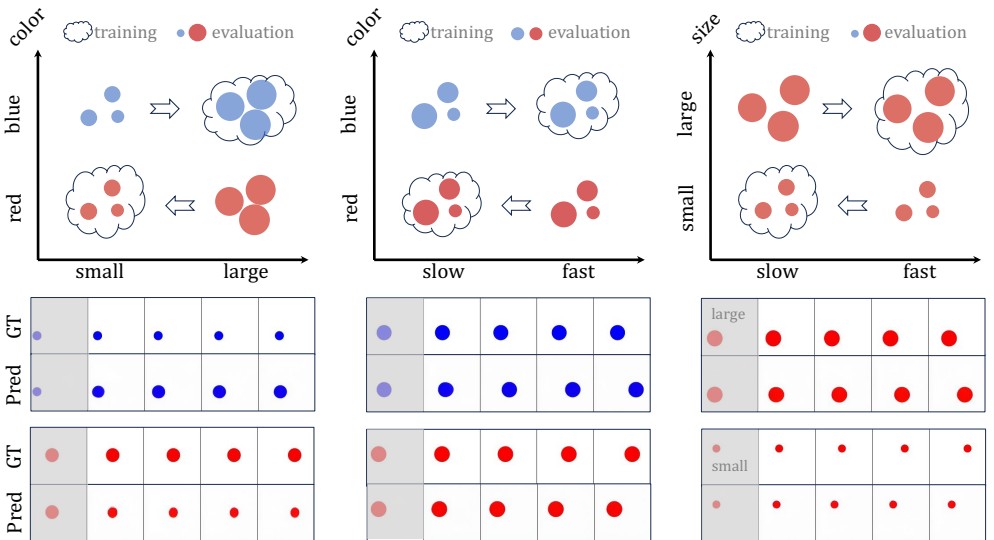

Figure 9: Uniform motion. (1) Color *v.s.* size, (2) Color *v.s.* velocity, (3) Size *v.s.* velocity. The change in velocity is evident when comparing the ground truth with the predicted videos.

**Attribute composition**. As shown in Figure 14 (1)-(2), certain attribute pairs—such as velocity and size, or color and size—exhibit some degree of combinatorial generalization.

**Spatial composition**. As given by Figure 11 (left side) in the appendix, the training data contains two distinct types of physical events. One type involves a blue square moving horizontally with a constant velocity while a red ball remains stationary. In contrast, the other type depicts a red ball moving toward and then bouncing off a wall while the blue square remains stationary. At test time, when the red ball and the blue square are moving simultaneously, the learned model is able to generate the scenario where the red ball bounces off the wall while the blue square continues its uniform motion.

**Temporal combination.** As illustrated on the right side of Figure 11, when the training data includes distinct physical events—half featuring two balls colliding without bouncing and the other half showing a red ball bouncing off a wall—the model learns to combine these events temporally. Consequently, during evaluation, when the balls collide near the wall, the model accurately predicts the collision and then determines that the blue ball will rebound off the wall with unchanged velocity.

With these attribute, spatial, and temporal combinatorial patterns, the video generation model can identify basic physical events in the training set and combine them across attributes, time, and space to generate videos featuring complex chains of physical events.

## 5.5 IS VIDEO SUFFICIENT FOR COMPLETE PHYSICS MODELING?

For a video generation model to function as a world model, the visual representation must provide sufficient information for complete physics modeling. In our experiments, we found that visual ambiguity leads to significant inaccuracies in fine-grained physics modeling. For example, in Figure 10, it is difficult to determine if a ball can pass through a gap based on vision alone when the size difference is at the pixel level, leading to visually plausible but incorrect results. Similarly, visual ambiguity in a ball's horizontal position relative to a block can result in different outcomes. These findings suggest that relying solely on visual representations, may be inadequate for accurate physics modeling.

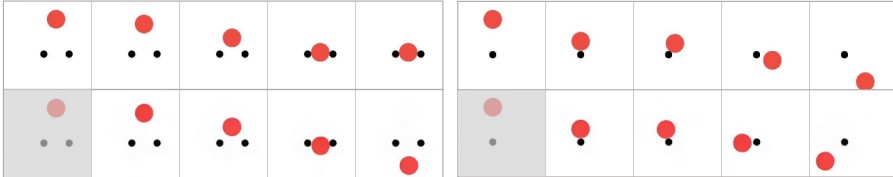

Figure 10: First row: Ground truth; second row: generated video. Ambiguities in visual representation result in inaccuracies in fine-grained physics modeling.

## 6 RELATED WORKS

**Video generation.** Open-set video generation is mostly based on diffusion models (Rombach et al., 2022; Ho et al., 2022b;a; He et al., 2024) or auto-regressive models (Yu et al., 2023a;b; Kondratyuk et al., 2023). These models often require a pretrained image or video VAE (Kingma, 2013; Van Den Oord et al., 2017) for data compression to improve computational efficiency. Some approaches leverage pretrained Text-to-Image (T2I) models for zero-shot (Khachatryan et al., 2023; Zhang et al., 2023b) or few-shot (Wu et al., 2023) video generation. Additionally, Image-to-Video(I2V) generation (Zeng et al., 2024; Girdhar et al., 2023; Zhang et al., 2023a) shows that video quality improves substantially when conditioned on an image. The Diffusion Transformer (DiT) (Peebles & Xie, 2023) demonstrates better scaling behavior than U-Net (Ronneberger et al., 2015) for T2I generation. Sora (Brooks et al., 2024) leverages the DiT architecture to directly operate on spacetime patches of video and image latent codes. Our model follows Sora's architecture and conceptually aligns with I2V generation, relying on image(s) for conditioning instead of text prompts.

**World model.** World models (Ha & Schmidhuber, 2018) aim to learn models that can accurately predict how an environment evolves after some actions are taken. Previously, they often operated in an abstracted space and were used in reinforcement learning (Sutton, 2018) to enable planning (Silver et al., 2016; Schrittwieser et al., 2020) or facilitate policy learning through virtual interactions (Hafner et al., 2019; 2020). With the advancement of generative models, world models can now directly work with visual observations by employing a general framework of conditioned video generation. For example, in autonomous driving (Hu et al., 2023; Jia et al., 2023; Gao et al., 2024; Zheng et al., 2024), the condition is the driver's operations, while in robot world models (Yang et al., 2023; Black et al., 2023; 1xw, 2024), the condition is often the control signals. Genie (Bruce et al., 2024) instead recovers the conditions from video games in an unsupervised learning manner. In our physical law discovery setting, it does not require a per-step action/condition since the physical event is determined by the underlying laws once an initial state is specified. See more related works in Appendix A.1.

## 7 CONCLUSION

Video generation is believed as a promising way towards scalable world models. However, its capability to learn physical laws from visual observations has not yet been verified. We conducted the first systematic study in this area by examining its generalization performance across three typical scenarios: in-distribution, out-of-distribution (OOD), and combinatorial generalization. The findings indicate that scaling alone cannot address the OOD problem, although it does enhance performance in other scenarios. Our in-depth analysis suggests that video model generalization relies more on referencing similar training examples rather than learning universal rules. We observed a prioritization order of color > size > velocity > shape in this "case-based" behavior. In conclusion, our study suggests that naively scaling is insufficient for video generation models to discover physical laws.

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

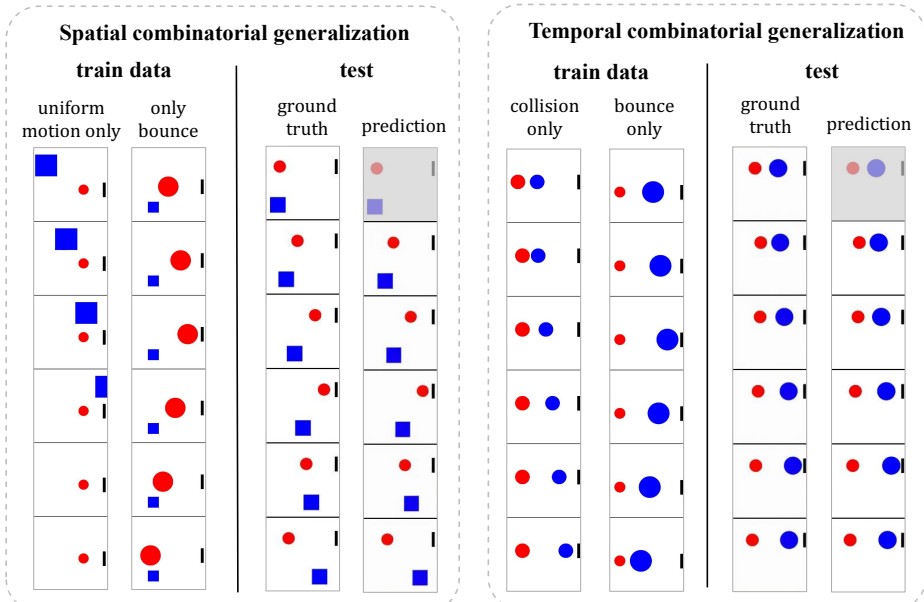

Figure 11: Spatial and temporal combinatorial generalization. The two subsets of the training set contain disjoint physical events. However, the trained model can combine these two types of events across spatial and temporal dimensions.

# A  APPENDIX

## A.1  MORE RELATED WORKS

**Physical reasoning.** It refers to the ability to understand and predict the way objects will interact under certain conditions according to physical laws. Melnik et al. (2023) categorize physical reasoning tasks into *passive* and *interactive* tasks. Passive tasks often require the AI to predict certain properties of objects, *e.g.*, materials (Bell et al., 2015; Bouman et al., 2013), physical parameters (Wu et al., 2016), stability of a physical system (Groth et al., 2018), or involve question-answering for the agent to recognize conceptual differences (Weitnauer et al., 2023; Yi et al., 2019), or describe a physical scenario (Ates et al., 2020). For interactive tasks, AI is required to control some objects in the environment to complete certain tasks based on physical commonsense, *e.g.*, solving classical mechanics puzzles (Bakhtin et al., 2019), flying a bird to reach a target position (Xue et al., 2023), and using a tool (Allen et al., 2020). Two works closely related to ours are Girdhar et al. (2020) and de Silva et al. (2020). Girdhar et al. (2020) introduce a forward prediction model to aid physical reasoning but do not address what is learned in the prediction model. de Silva et al. (2020) attempt to discover universal physical laws from data with abstracted internal states and human expertise introduced in the dynamic model design. In contrast, we focus on recovering physical laws from raw observation without any human priors, akin to a newborn baby.

## A.2  DIFFUSION PRELIMINARIES

Let $p(x)$ be the real data distribution. Diffusion models (Ho et al., 2020; Song et al., 2020) learn the data distribution by denoising samples from a noise distribution step-by-step. In this paper, we use the Gaussian diffusion models, where the video $V$ is progressively corrupted by gaussian noise $\epsilon \sim \mathcal{N}(0, \boldsymbol{I})$ during the forward process, denoted by

$$V_t = \alpha_t V + \beta_t \epsilon, \tag{1}$$

where $\alpha_t, \beta_t$ are the time-dependent noise scheduler. We use the original DDPM formulation (Ho et al., 2020), where $\alpha_t = \sqrt{\gamma_t}, \beta_t = \sqrt{1 - \gamma_t}$, $\gamma_t$ is a monotonically decreasing scheduler from 1 to 0. The diffusion models are trained to reverse the forward corruptions, denoted by

$$\mathbb{E}_{V \sim p(x), t \sim \mathcal{U}(0, \boldsymbol{I}), \epsilon \sim \mathcal{N}(0, \boldsymbol{I})} \left[ \|\mathbf{y} - p_\theta(V_t, \mathbf{c}, t)\|^2 \right], \tag{2}$$

where the target $\mathbf{y}$ can be the corrupted noise $\epsilon$, the original video $V$, or the velocity between data and noise. Following Salimans & Ho (2022), we use the velocity prediction to train the video diffusion models, denoted by

$$\mathbf{y} = \sqrt{1 - \gamma_t}\epsilon - \sqrt{\gamma_t}V. \tag{3}$$

Also, to eliminate the training and inference gap and ensure a zero signal-to-noise ratio at the final timestep, following Lin et al. (2024), we set $\gamma_t$ to 1 when $t = 1$.

### A.3 LATENT VIDEO DIFFUSION MODEL

#### A.3.1 VAE ARCHITECTURE AND PRETRAIN

We commence with the structure of the SD1.5-VAE, retaining the majority of the original 2D convolution, group normalization, and attention mechanisms on the spatial dimensions. To inflate this structure into a spatial-temporal auto-encoder, we convert the final few 2D downsample blocks of the encoder and the initial few 2D upsample blocks of the decoder into 3D ones, and employ multiple extra 1D layers to enhance temporal modeling. For all the downsample blocks where temporal downsampling is required, we replace all the original 2D downsample layers with re-initialized causal 3D downsample layers by adding causal paddings to the head of the frame sequence (Yu et al., 2023b) and introduce an additional causal 1D convolution layer after the original *ResNetBlock*. As for the decoder part, all the 2D *Nearest-Interpolation* operations are substituted by a 2D convolution layer and a channel-to-space transformation, which are specifically initialized to behave precisely the same as *Nearest-Interpolation* operations before the first training step. For the discriminator part, we inherit the structure of the original 2D PatchGAN (Isola et al., 2016) discriminator used by SD1.5-VAE, and design a 3D PatchGAN discriminator based on the 2D version. Different from the generator module, we train the group of discriminators from scratch for the consideration of stability. Subsequently, all parameters of the (2+1)D-VAE are jointly trained with high-quality image and video data to preserve the capability of appearance modeling and to enable motion modeling. For the image dataset, we filter data samples from LAION-Aesthetics (Beaumont & Schuhmann, 2022), COYO (Byeon et al., 2022) and DataComp (Gadre et al., 2023) with high aesthetics and clarity to form a high-quality subset. As for the video dataset, we collect a high-quality subset from Vimeo-90K (Xue et al., 2017), Panda-70M (Wang et al., 2020) and HDVG Wang et al. (2023). In the training process, we train the entire structure for 1M steps and only the random resized crop and random horizontal flip are applied in the data augmentation process.

#### A.3.2 VAE RECONSTRUCTION

In this paper, we *fix* the pretrained VAE encoder and use it as a video compressor. To verify the VAE's ability to accurately encode and decode the physical event videos, we evaluate its reconstruction performance. Specifically, we use the VAE to encode and decode (i.e., reconstruct) the ground truth videos and calculate the reconstruction error, $e_{\text{recon}}$. We then compare this error to the ground truth error, $e_{\text{gt}}$, as shown in Table 3. The results show that $e_{\text{recon}}$ is very close to $e_{\text{gt}}$, and both are an order of magnitude lower than the OOD error, as illustrated in Figure 3. It confirms the pretrained VAE's ability to accurately encode and decode the physical event videos used in this paper.

Table 3: Comparison of errors for ground truth videos and VAE reconstruction videos.

| Scenario | Ground Truth Error | VAE Reconstruction Error |
|---|---|---|
| Uniform Motion | 0.0099 | 0.0105 |
| Collision | 0.0117 | 0.0131 |
| Parabola | 0.0210 | 0.0212 |

#### A.3.3 DIT IMPLEMENTATION DETAILS

Following Sora (Brooks et al., 2024), we directly use the DiT architecture (Peebles & Xie, 2023) as the denoising model. The only modification is that we employ a 3D variant of RoPE for better fitting the video data. Throughout the paper, we use a peak learning rate of $1 \times 10^{-4}$ with cosine decay, a bach size of 256. All models are trained with the AdamW (Diederik & Ba, 2015; Loshchilov,

2017) optimizer, with $\beta_1 = 0.9$, $\beta_2 = 0.999$ and the weight decay weight equals to $0.01$. Horizontal flipping is the only data augmentation if not specified.

## A.4 DETAILED EXPERIMENTAL SETUP

### A.4.1 FUNDAMENTAL PHYSICAL SCENARIOS DATA

For the Box2D simulator, we initialize the world as a $10 \times 10$ grid, with a timestep of $0.1$ seconds, resulting in a total time span of $3.2$ seconds ($32$ frames). For all scenarios, we set the radius $r \in [0.7, 1.5]$ and velocity $v \in [1, 4]$ as in-distribution (in-dist) ranges. Out-of-distribution (OOD) ranges are defined as $r \in [0.3, 0.6] \cup [1.5, 2.0]$ and $v \in [0, 0.8] \cup [4.5, 6.0]$.

**Collision Scenario:** The four degrees of freedom (DoFs) are the masses of the two balls and their initial velocities, fully determining the collision outcomes. We generate 3k, 30k, and 3M training samples by sampling grid points from the 4-dimensional in-dist joint space of radii and velocities. For in-dist evaluation, we randomly sample about 2k points from the grid, ensuring they are not part of the training set. For OOD evaluation, we sample from the OOD ranges, generating approximately 4.8k samples across six OOD levels: (1) only $r_1$ OOD, (2) only $v_1$ OOD, (3) both $r_1$ and $r_2$ OOD, (4) both $v_1$ and $v_2$ OOD, (5) $r_1$ and $v_1$ OOD, and (6) $r_1$, $v_1$, $r_2$, and $v_2$ OOD. Additionally, for collisions, we ensure that all collisions occur after the 4th frame in each video, allowing the initial velocities of both balls to be inferred from the conditioned frames.

**Uniform and Parabolic Motion:** The two DoFs are the ball's mass and initial velocity. We generate 3k, 30k, and 3M training samples by sampling from the 2-dimensional in-dist joint space of radius and velocity. For in-dist evaluation, we sample approximately 1.05k for uniform motion and 1.1k for parabolic motion. For OOD evaluation, we generate about 2.4k (uniform motion) and 2.5k (parabolic motion) samples across three OOD levels: (1) only $r$ OOD, (2) only $v$ OOD, and (3) both $r$ and $v$ OOD.

### A.4.2 COMBINATORIAL EXPERIMENT EVALUATION SETUP

To evaluate the generated videos, we conducted human evaluations to determine the abnormal ratio, defined as the proportion of videos that violate physical laws as assessed by human intuition. We utilized 60 templates for training and reserved 10 unused templates for evaluation. For the in-template evaluation set, we sampled 2 videos from each of the 60 training templates, forming a total of 120 videos (excluded from training). For the out-of-template evaluation set, we sampled 10 videos from each of the 10 reserved templates, forming a total of 100 videos. We recruited 10 human evaluators to assess the videos. Each evaluator independently reviewed all videos and determined whether the physical processes depicted adhered to physical laws. We average the abnormal rates over all evaluators and report the result. Each evaluator received the following instruction: "*For each video shown to you, all objects start with free fall. You should judge whether the physical processes in the video obey physical laws based on your intuition. Select 'Yes' if the process adheres to physical laws and 'No' otherwise. Please make your judgment based on your first impression*".

## A.5 OTHER SOTA VIDEO GENERATION MODELS

In this section, we explore how the findings in this paper apply to open-source pretrained models, such as CogVideo (Hong et al., 2022) and Stable Video Diffusion (SVD) (Blattmann et al., 2023).

### A.5.1 IS THE PRIORITIZATION RELEVANT TO VAE?

To evaluate how much the prioritization order color > size > velocity > shape is influenced by the explicit instantiation of the video VAE used in our work, we perform experiments with alternative architectures. Specifically, we replace the VAE from our primary experiments with the VAE from the recently released CoGVideo and rerun the tests. By freezing the CoGVideo VAE and training only the diffusion component, we find that the prioritization order remains consistent.

As detailed in Section 5.3, we designed experiments to compare the prioritization of velocity and shape. Videos of balls with low velocities and squares with high velocities were used for training. During testing, we evaluated scenarios where a ball with high velocity transformed into a square

immediately after the conditioned frames, and a square with low velocity transitioned into a ball. Across 1,400 test cases, no exceptions were observed. The results are identical to those presented in Figure 8 (3), confirming that the prioritization order velocity > shape holds. For velocity and size, as shown in Figure 12 (1), the model consistently maintained the initial size and velocity for most test cases, even beyond the training distribution. However, a slight preference for size over velocity was observed, particularly when radius and velocity values were at extreme ends (top-left of the plot). These results confirm that size holds a higher priority than velocity. Finally, for the prioritization between color and velocity, we trained the model on high-speed blue balls and low-speed red balls. During testing, high-speed red balls appeared much slower than their conditioned velocity, while no balls in the test set changed their color. As illustrated in Figure 12 (2), this demonstrates that color is prioritized over velocity. In summary, our experiments with the CoGVideo VAE confirm the robustness of the prioritization order color > size > velocity > shape, demonstrating that these findings generalize across different VAE pretrain.

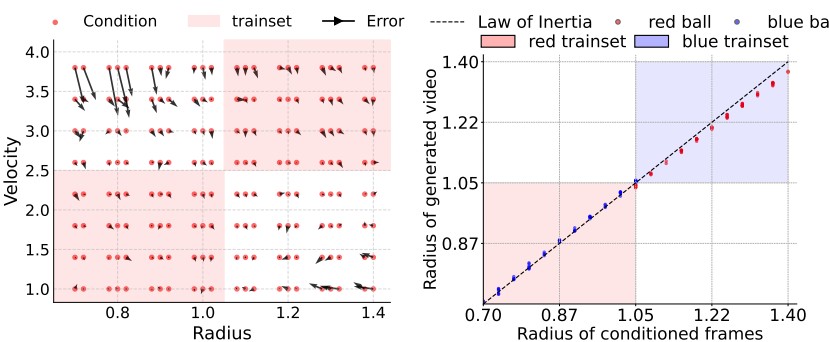

Figure 12: Prioritization experiments with CogVideo's VAE. (1) Velocity *v.s.* size. (2) Color *v.s.* velocity.

### A.5.2 CAN PRETRAINED MODELS LEARN PHYSICAL LAWS?

The primary goal of this paper is to answer the fundamental question: "Can a video generation model learn physical laws?" Existing pretrained models are built on large-scale datasets sourced from publicly available internet content. However, the extent to which these datasets contain videos about physical laws or structured physical interactions remains unknown. This lack of transparency leads to critical concern - contamination of evaluation data: There is a risk that pretrained models may have been exposed to data similar to our evaluation experiments during training. This potential overlap undermines the validity of assessing their ability to learn physical reasoning. To address these concerns, we chose to train a model from scratch, which provides complete control over the training data. This approach allows us to precisely ensure that the results stem from the model's actual learning process rather than potential pretraining data overlap or contamination. This methodology is supported by recent research into the mechanisms of language models, where training relatively small models from scratch has proven effective in isolating and analyzing reasoning processes more rigorously.

To assess whether the inability to learn physical laws is specific to our setup or extends to open-source pretrained models, we conducted experiments using the Stable Video Diffusion (SVD) model on a representative uniform motion task. We fine-tuned SVD for 300k steps, following the same training setup as DiT-B in our experiments. The results, summarized in Table 4, reveal several key observations. First, SVD's VAE demonstrates a very small reconstruction error, indicating that its VAE component effectively preserves kinematic information. However, the diffusion component of SVD, despite pretraining, performs worse than DiT-B in both in-distribution (ID) and out-of-distribution (OOD) predictions. We attribute this to the considerable gap between the pretraining dataset (internet videos) and the simplified kinematic test scenarios used in this evaluation. Notably, the OOD error for SVD is an order of magnitude larger than its ID error, a trend consistent with DiT-B. This finding reinforces the robustness of our conclusion that pretraining alone does not address the inability of video generation models to generalize to OOD scenarios, a limitation that persists across architectures, including open-source pretrained models like SVD.

Table 4: Results of finetuning SVD model.

| Model | ID Error | OOD Error |
|---|---|---|
| GT | 0.0099 | 0.0104 |
| DiT-B (from scratch, ours) | 0.0138 | 0.3583 |
| SVD-VAE-Recon | 0.0103 | 0.0107 |
| SVD-Finetune | 0.0505 | 0.9081 |

## A.6 MORE EXPERIMENTS AND DISCUSSIONS

### A.6.1 CAN LANGUAGE AND NUMERICS AID IN LEARNING PHYSICAL LAWS?

As discussed in Section 3, video generation based solely on image frames fails to learn physical laws, showing significant prediction errors in OOD scenarios, despite containing all the necessary information. In reinforcement learning, numerical values (*e.g.,* states) are often used as conditions for world models (Hafner et al., 2023), and language representations have shown generalization capabilities in LLMs (Riveland & Pouget, 2024). This raises the question: can additional multimodal inputs, such as numerics and text, improve video prediction and capture physical laws? We experimented with collision scenarios and DiT-B models, adding two variants: one conditioned on vision and numerics, and the other on vision and text. For numeric conditioning, we map the state vectors to embeddings and add the layer-wise features to video tokens. For text, we converted initial physical states into natural language descriptions, obtained text embeddings using a T5 encoder, and then add a cross-attention layer to aggregate textual representations for video tokens.

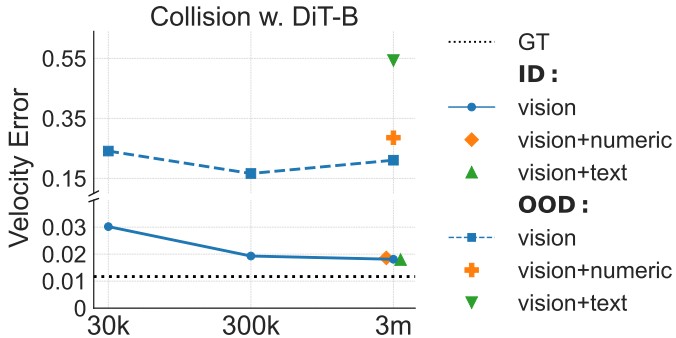

Figure 13: Comparison of different modal conditions for video generation.

As shown in Figure 13, for in-distribution generalization, adding numeric and text conditions resulted in prediction errors comparable to using vision alone. This suggests that visual frames already contain sufficient information for accurate predictions, and the additional numeric or text data do not provide further benefits. However, in OOD scenarios, the vision-plus-numerics condition exhibited slightly higher errors, while the vision-plus-language condition showed significantly higher errors. We hypothesize that the embeddings of language tokens and numerics may cause the model to overfit to specific patterns present in the training data, thereby impairing its ability to generalize to unseen OOD scenarios. Additionally, language tokens are discrete and exhibit greater variability compared to continuous numeric embeddings, making the model more susceptible to overfitting when using language inputs, accounting for the higher OOD errors in the vision-plus-language condition compared to the vision-plus-numerics condition.

### A.6.2 CONTINUOUS EXPERIMENT FOR PAIRWISE COMPARISON

We also conducted attribute comparison experiments using continuous values for velocity and size to quantify these prioritizations. For velocity *vs.* size, the combinatorial generalization performance is surprisingly good. The model effectively maintains the initial size and velocity for most test cases beyond the training distribution. However, a slight preference for size over velocity is noted, particularly with extreme radius and velocity values (top left and bottom right in Figure 14 (1)). In

Figure 14 (2), color can be combined with size most of the time. Conversely, for color *vs.* velocity in Figure 14 (3), high-speed blue balls and low-speed red balls are used for training. At test time, low-speed blue balls appear much faster than their conditioned velocity. No ball in the testing set changes its color, indicating that color is prioritized over velocity. The above analysis is consistent with the conclusion drawn from the binary attribute comparisons: color > size > velocity > shape.

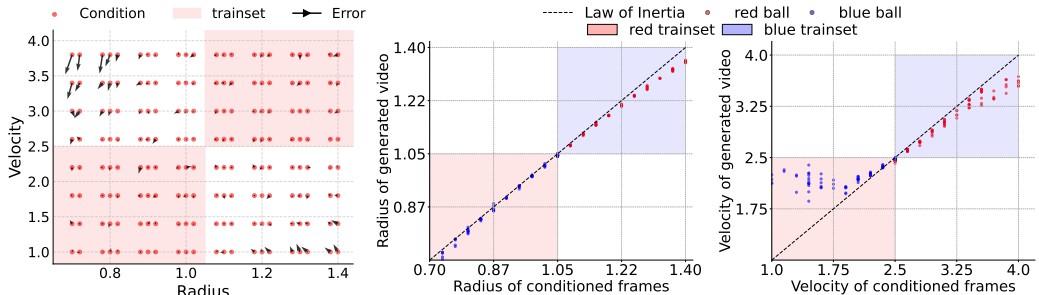

Figure 14: Uniform motion. (1) Velocity *v.s.* size: The arrow → indicates the direction of generated videos shifting from their initial conditions. (2) Color *v.s.* size: Models are trained with small red balls and large blue balls, and evaluated on reversed color-size pair conditions. All generated videos retain the initial color but show slight size shifts from the original. (3) Color *v.s.* velocity: Models are trained with low-speed red balls and high-speed blue balls, and evaluated on reversed color-velocity pair conditions. All generated videos retain the initial color but show large velocity shifts from the original.

To confirm that the shift observed in Figure 14 is indeed due to an attribute prioritization and not due to general behavior, we performed control experiments corresponding to the main experiments shown in panels (2) and (3) of Figure 14. These experiments were designed without a attribute conflict (i.e., only the red color was used). As shown in Figure 15, even in extreme cases, the predictions remained accurate without any change. This result indicates that the drift observed in Figure 14 arises specifically from the attribute conflict and the presence of attribute prioritization.

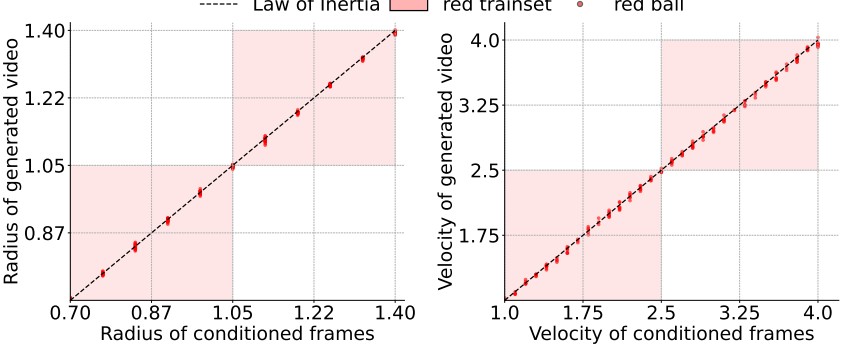

Figure 15: Control experiment for pairwise comparison. These experiments were designed without a attribute conflict (i.e., only the red color).

### A.6.3 PRINCIPLE BEHIND DATA RETRIEVAL IN THE DIFFUSION MODEL

As discussed in Section 5.3, the diffusion model appears to rely more on memorization and case-based imitation, rather than abstracting universal rules for OOD generalization. The model exhibits a preference for specific attributes during case matching, with a prioritization order of color > size > velocity > shape. In this section, we aim to explore the reasoning behind this prioritization.

Since the diffusion model is trained by minimizing the loss associated with predicting VAE latent, we hypothesize that the prioritization may be related to the distance in VAE latent space (though we use pixel space here for clearer illustration) between the test conditions and the training set. Intuitively, when comparing color and shape as in Figure 14 (1), a shape change from a ball to a rectangle results in minor pixel variation, primarily at the corners. In contrast, a color change from blue to red causes a more significant pixel difference. Thus, the model tends to preserve color while allowing shape to

vary. From the perspective of pixel variation, the prioritization of color > size > velocity > shape can be explained by the extent of pixel change associated with each attribute. Changes in color typically result in large pixel variations because it affects nearly every pixel across its surface. In contrast, changes in size modify the number of pixels but do not drastically alter the individual pixels' values. Velocity affects pixel positions over time, leading to moderate variation as the object shifts, while shape changes often involve only localized pixel adjustments, such as at edges or corners. Therefore, the model prioritizes color because it causes the most significant pixel changes, while shape changes are less impactful in terms of pixel variation.

To further validate this hypothesis, we designed a variant experiment comparing color and shape, as shown in Figure 16. In this case, we use a blue ball and a red ring. For the ring to transform into the ball without changing color, it would need to remove the ring's external color, turning it into blank space, and then fill the internal blank space with the ball's color, resulting in significant pixel variation. Interestingly, in this scenario, unlike the previous experiments shown in Figure 14 (1), the prioritization of color > shape does not hold. The red ring can transform into either a red ball or a blue ring, as demonstrated by the examples. This observation suggests that the model's prioritization may indeed depend on the complexity of the pixel transformations required for each attribute change. Future work could explore more precise measurements of these variations in pixel or VAE latent space to better understand the model's training data retrieval process.

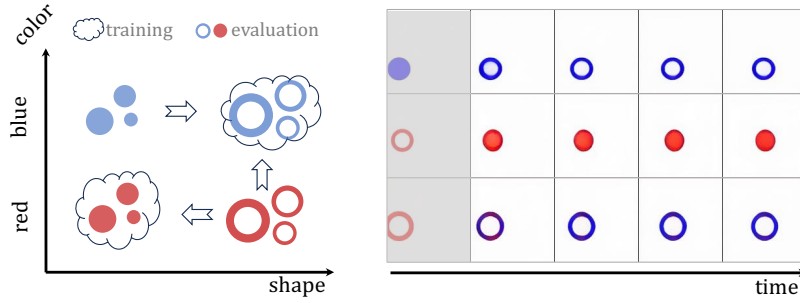

Figure 16: Uniform motion. Color vs. shape. The shapes are a ball and a ring. Transforming from a ring to a ball leads to a large pixel variation.

### A.6.4 FAILURE CASES IN COMBINATORIAL GENERALIZATION

In Figure 11, we present successful cases of spatial and temporal combinatorial generalization, where the trained model can combine these two types of events across spatial and temporal dimensions to generate videos under unseen conditions. However, the model does not always succeed in performing such compositions, and here we illustrate some failure cases. As shown in Figure 17, when the training set lacks a red ball in a bounce event, the model *sometimes* struggles. In the model's generated video, the red ball sometimes disappears after a collision. This likely stems from the model's data retrieval mechanism: since the training set does not include a red ball in a collision scenario, when collision happens, the model retrieves similar training cases without the red ball, causing it to vanish post-collision. In summary, while combinatorial generalization allows the diffusion model to generate novel videos by composing spatial and temporal segments from the training set, its reliance on data retrieval limits its effectiveness. As a result, the model may produce unrealistic outcomes by retrieving and combining segments without understanding the underlying rules.

### A.7 COMPARISON WITH ID/OOD GENERALIZATION WORKS

ID/OOD generalization Schott et al. (2021) in machine learning has been extensively discussed in foundational works. These discussions highlight the fundamental challenges of OOD generalization in machine learning. However, when applied to specific settings with unique data forms, distributions, and model inductive biases, it exhibits nuanced behaviors that remain scientifically significant. For instance, similar debates have emerged in the study of large language models (LLMs), where considerable attention has been devoted to determining whether models learn underlying rules or rely on case-based memorization for performing arithmetic (Hu et al., 2024). These insights have spurred

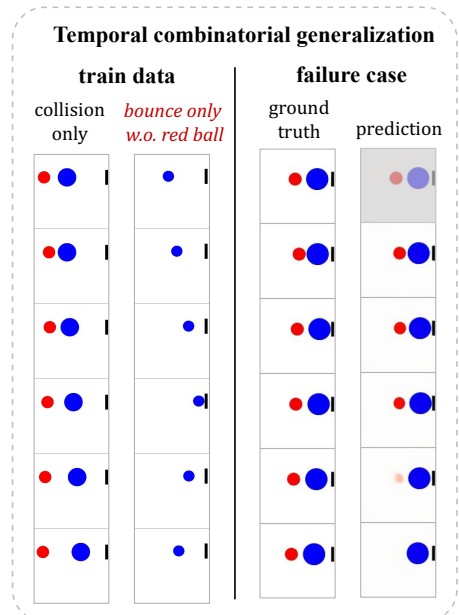

Figure 17: Failure cases in combinatorial generalization. Note that the bounce cases in the training set do not include the red ball.

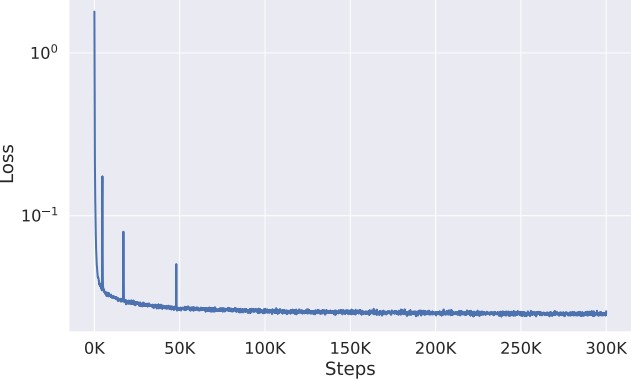

Figure 18: Training loss in Equation (2) curve of DiT-L model trained on 3M collision videos.

advancements in rule-based reasoning approaches within the LLM community. Similarly, our work investigates how video generation models handle generalization within their own domain.

Our findings go beyond the simplistic observation that ID scenarios succeed while OOD scenarios fail. Instead, we provide deeper insights and discoveries specific to video generation:

1. **Addressing the Debate on Learning Physical Laws in Video Generation Models:** Through systematic experiments, we provide a clear answer to whether physical laws can be learned by scaling video generation models. This has been a topic of debate in the video generation community. For instance, OpenAI's Sora Technical Report (Brooks et al., 2024) suggests that video generation models can learn rules and act as world simulators, implying they might generalize universal laws. Our findings challenge this assumption, showing that current video generation models fail to learn physical laws as universal rules.

2. **Scaling Guidance for Combinatorial Generalization:** We demonstrate the importance of combinatorial generalization and identify scaling laws for improving generalization in video generation models. Our findings highlight that increasing the diversity of combinations in the training data is more effective for achieving realistic physics than merely scaling the data volume or model size. This offers actionable guidance for building video generation models that better capture physical realism.

3. **Revealing the Generalization Mechanism and Understanding the Boundaries of Video Generation Models:** We uncover how video generation models generalize, primarily relying on referencing similar training examples rather than learning underlying universal principles. This provides a deeper understanding of their limitations and biases in representing physical phenomena.

These findings transcend traditional ID/OOD analyses, offering a more detailed understanding of the limitations and potential of video generation models.

## A.8 VISUALIZATION

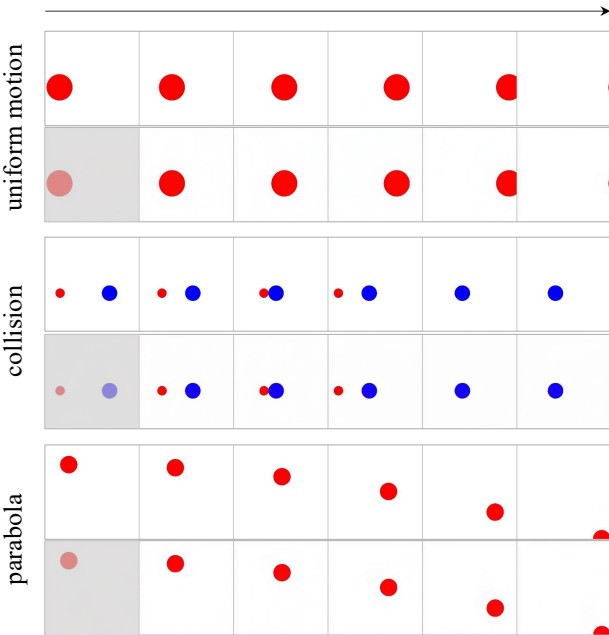

Figure 19: The visualization of *in-distribution evaluation* cases with very *small* prediction errors.

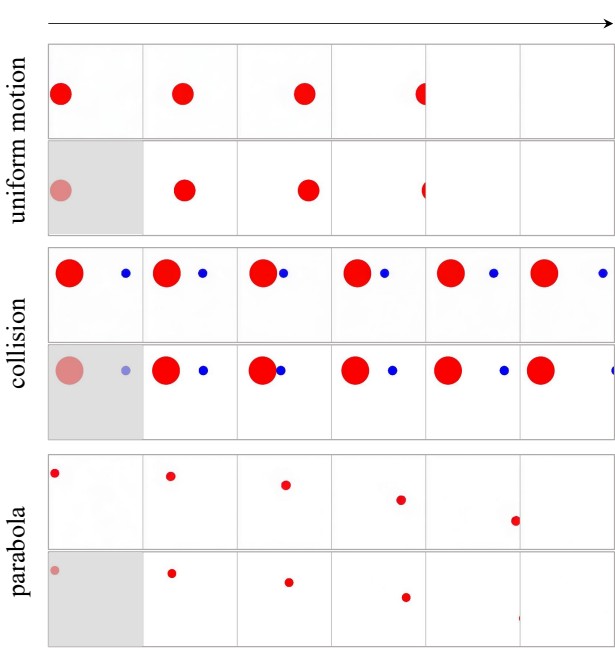

Figure 20: The visualization of *out-of-distribution evaluation* cases with *large* prediction errors.

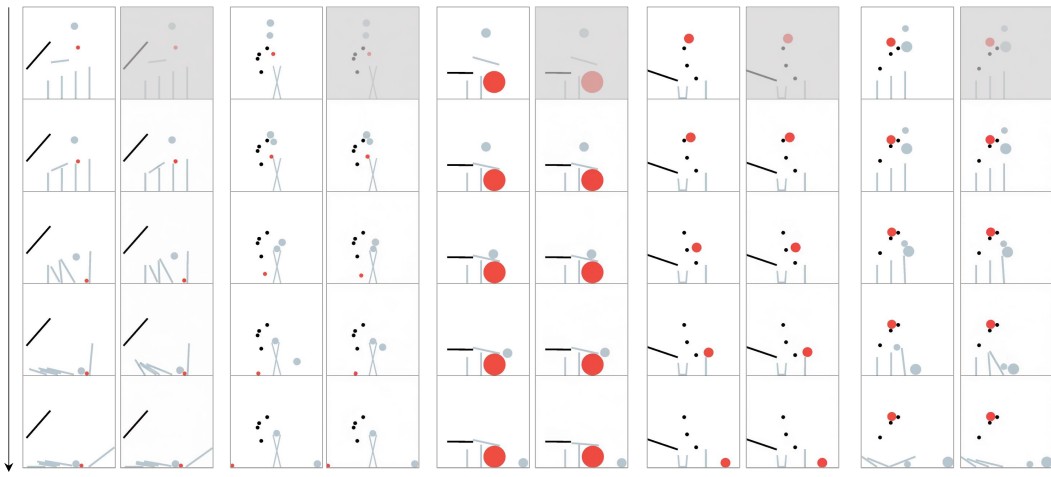

Figure 21: The visualization of *out-of-template evaluation* cases that appear *plausible and adhere to physical laws*, generated by DiT-XL trained on 6M data (60 templates). Zoom in for details. Notably, the first four cases generated by the model are nearly identical to the ground truth. In some cases, such as the rightmost example, the generated video seems physically plausible but differs from the ground truth due to visual ambiguity, as discussed in Section 5.5.

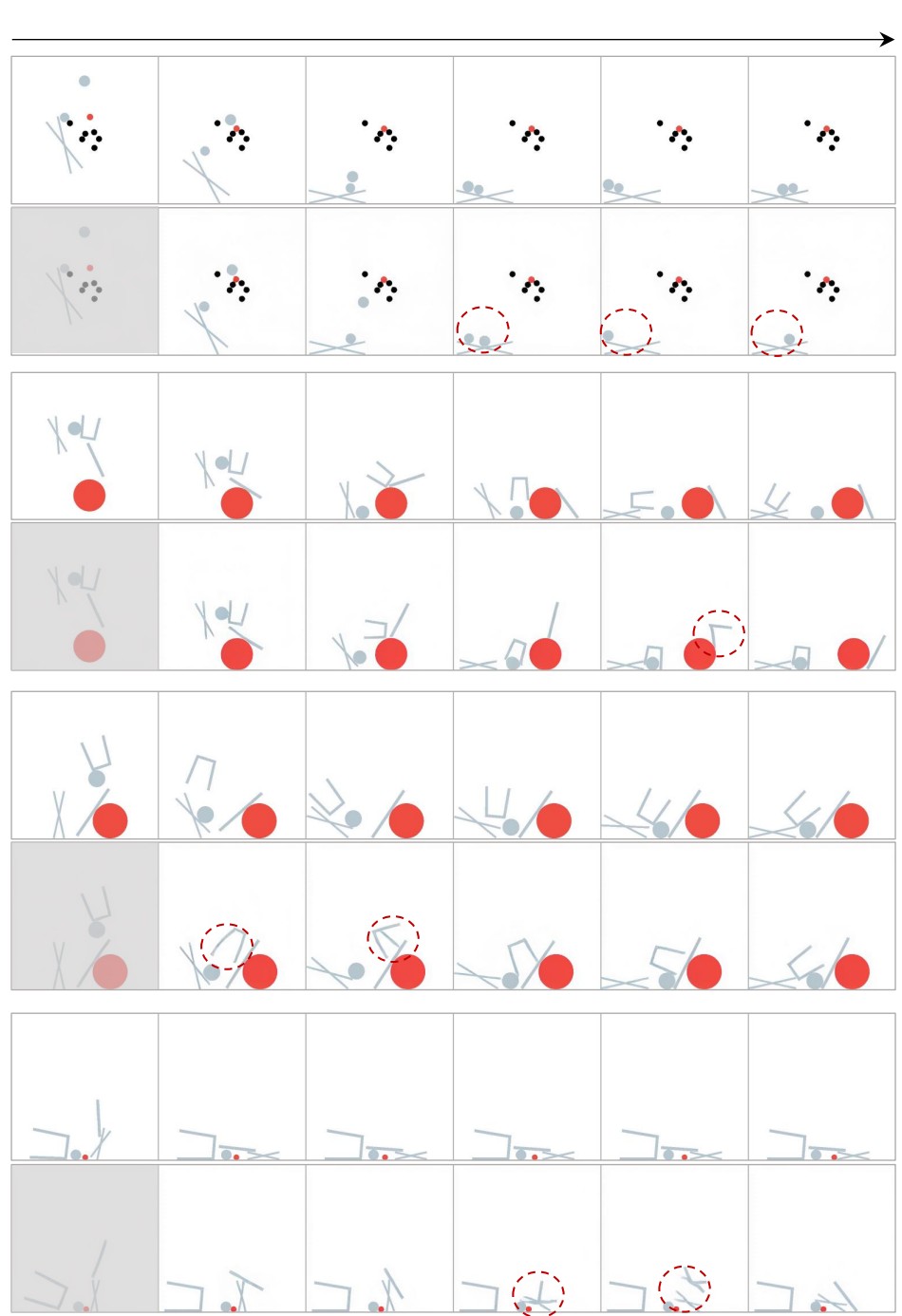

Figure 22: The visualization of *out-of-template evaluation* cases that appear *abnormal and violate physical laws*, generated by DiT-XL trained on 6M data (60 templates). Abnormalities are highlighted with red dotted circles. **Case 1**: A grey ball suddenly disappears. **Case 2**: The rigid-body bar breaks in several intermediate frames during contact with the ball, then recovers after contact. **Case 3**: The rigid-body jar fails to maintain its shape when interacting with the bar in several intermediate frames. **Case 4**: The rigid-body bar breaks in several intermediate frames during contact with the standing sticker. Most of the abnormal cases we observed involve object disappearance or shape inconsistencies, which can be explained by the case matching preference discussed in Section 5.3.

