# OpenReview forum: "How Far Is Video Generation from World Model: A Physical Law Perspective"
_ICLR.cc/2025/Conference — Submitted to ICLR 2025_

### Official Review · Reviewer_r3mF · 2024-10-27

**Soundness:** 3
**Presentation:** 3
**Contribution:** 2
**Rating:** 3
**Confidence:** 5

**Summary:**

This paper examines video generation models' ability to learn physical laws from visual data, testing their generalization across in-distribution (ID), out-of-distribution (OOD), and combinatorial scenarios. Claimed key contributions include:
1. Evaluation Framework: A structured assessment for how well models follow physical laws across different data distributions.
2. Scaling Insights: While data scaling improves ID generalization, it has minimal effect on OOD.
3. Prioritization in Generalization: Models prioritize attributes (color > size > velocity > shape) when referencing training data.
These findings could provide some insights for building robust world models in video generation.

**Strengths:**

The paper presents limited originality, as it largely builds on existing approaches to video generation and world modeling, applying these to well-studied physical scenarios without introducing novel methodologies or frameworks. (see weaknesses)

However, it demonstrates good quality in experimental design and data scaling, using systematic evaluations across in-distribution, out-of-distribution, and combinatorial scenarios, which provides valuable insights into model limitations.

The **clarity of the paper is strong**, with well-defined sections on methodology, findings, and practical insights, making the results and their implications easily accessible to the reader. In terms of **significance**, the contributions are moderate, as the findings—while informative—are more incremental than groundbreaking. They highlight known limitations in model generalization rather than offering transformative advances for video generation or physical law discovery.

Overall, the paper is a solid study with strengths in clarity and quality but limited novelty and moderate significance in its contributions to the field.

**Weaknesses:**

The paper's main weaknesses revolve around its misalignment between its stated focus on physical law discovery and the actual content of its experiments:

1. Although the paper claims to investigate the learning of physical laws, the experiments primarily assess model sensitivity to superficial attributes like color, size, and shape. This focus on object properties, rather than fundamental dynamics such as force or acceleration, detracts from the paper’s relevance to physical law discovery, as these experiments fail to represent deeper physical principles.

2.  The findings are primarily reiterations of known phenomena—such as the ease of in-distribution (ID) generalization versus the challenges of out-of-distribution (OOD) generalization. These are well-established issues in machine learning, and the paper does not contribute fresh insights specific to physical law generalization.

3. Most observations (e.g., the prioritization of color over shape) apply to general video model generalization rather than to any specific domain of physics. Consequently, the paper’s insights into generalization mechanisms are not uniquely tied to physical laws, making the connection to physical modeling somewhat tenuous.

4.  The paper appears to position itself as a study on physical law discovery, yet the experimental design and findings align more closely with a standard exploration of OOD and ID generalization in video generation. This disconnect may suggest an attempt to frame familiar findings in a new light, without sufficiently novel contributions to the study of physical law generalization.

Overall, the paper does not fully deliver on its premise of exploring physical laws, focusing instead on known generalization issues in video models without advancing the state of research in physics-informed AI.

**Questions:**

Have you compared your findings with a long list of previous works on examining OOD and IND generalization of neural network models in general? For example, this paper:
Visual Representation Learning Does Not Generalize Strongly Within the Same Domain

---

> ### Author Response · Authors · 2024-11-22
>
> Before addressing the detailed questions, we would like to **clarify the position and overall goal of our paper**, as we noticed a misunderstanding (evident from the following quotations), likely due to our presentation.
>
> - "the paper does not contribute fresh insights specific to physical law generalization."
> - "The paper appears to position itself as a study on physical law discovery, "
> - "the paper does not fully deliver on its premise of exploring physical laws"
>
> The paper presents an empirical study on **whether scaling video generation can learn physical laws from video data**, as emphasized in the abstract and introduction. The primary motivation is the success of Sora demonstrating that video generation is promising in building physical world models, which requires verification.
>
> There are three key points worth noting.
> - First, we specifically focus on video generation models instead of other methods, meaning only video data is considered.
> - Second, we scale the data and models to ensure reliable conclusions.
> - **Third, we investigate the effectiveness of video generation models in recovering physical laws from data, rather than developing intellectual methods to discover new physical laws.** Therefore, we choose well-established physical laws to build our testbed. Discovering new physical laws with AI is a highly challenging topic and would benefit from the joint efforts of both AI and physics communities.
>
> We believe our study provides unique insights into understanding video generation models in learning physical laws.
> - We are conducting the first systematic study on this topic.
> - We present insightful conclusions for three generalization scenarios under scaling, which is crucial for the success of modern large models like Sora and has been neglected by traditional ML.
> - We explore the generalization mechanism of video generation models, revealing that these models fail to abstract general physical rules and instead mimic the closest training example, supported by clear experiments (Sec. 5.2).
> - We are the first to show that video generation models prioritize different factors when referencing training data: color > size > velocity > shape.
>
> We sincerely hope the reviewer will take a closer look at our paper and re-examine it based on our clarification. We will revise the paper to avoid potential misunderstandings. If there are any further questions or concerns, please feel free to contact us immediately.

---

> > ### Author Response · Authors · 2024-11-22
> >
> > `The paper's main weaknesses revolve around its misalignment between its stated focus on physical law discovery and the actual content of its experiments: Although the paper claims to investigate the learning of physical laws, the experiments primarily assess model sensitivity to superficial attributes like color, size, and shape.`
> >
> > Thank you for your feedback. Below, we clarify the focus and structure of our experiments:
> >
> > 1. **Main Experiments Focus on Physics, Not Superficial Attributes**: Contrary to the claim that our experiments primarily assess sensitivity to superficial attributes like color, size, and shape, our core experiments are designed to evaluate adherence to fundamental physical principles:
> > - Experiment 1 (ID/OOD Tests): Evaluates the model’s ability to **adhere to momentum conservation, energy conservation, and Newton’s laws of motion**, using **velocity error** as the primary metric. These tests focus solely on physical laws and do **not involve attributes such as color or shape.**
> > - Experiment 2 (Combinatorial Generalization): Measures **the abnormal rate of events that break physical laws**, assessing the model’s capacity to generalize and maintain physical consistency across diverse scenarios, not involving color or shape as well.
> >
> > These experiments are explicitly centered on **physical principles, not superficial object properties,** to rigorously evaluate the models’ grasp of physical laws.
> >
> > 2. **Exploratory Analysis on What Models Actually Learn**: After the main experiments showed that models fail to learn physical laws, we conducted analysis experiments to investigate what the model actually learned. These exploratory tests employed simpler setups involving color, size, and shape to vividly illustrate that the models rely on case-based memorization. These diagnostic experiments were supplementary.
> >
> > We hope this explanation clarifies the structure and intent of our work, demonstrating the relevance of our experiments to physical law.
> >
> > `The findings are primarily reiterations of known phenomena—such as the ease of in-distribution (ID) generalization versus the challenges of out-of-distribution (OOD) generalization. The paper does not contribute fresh insights specific to physical law generalization. `
> >
> > Thank you for your thoughtful feedback. We acknowledge that ID/OOD generalization in machine learning has been extensively discussed in foundational works. These discussions highlight the fundamental challenges of OOD generalization in machine learning.
> >
> > However, when applied to specific settings with unique data forms, distributions, and model inductive biases, it exhibit nuanced behaviors that remain scientifically significant. For instance, similar debates have emerged in the study of large language models (LLMs), where considerable attention has been devoted to determining whether models learn underlying rules or rely on case-based memorization for performing arithmetic [1]. These insights have spurred advancements in rule-based reasoning approaches within the LLM community. Similarly, our work investigates how video generation models handle generalization within their own domain.
> >
> > Our findings go beyond the simplistic observation that ID scenarios succeed while OOD scenarios fail. Instead, we provide deeper insights and discoveries specific to video generation:
> > 1. Through systematic experiments, we provide a clear answer to whether physical laws can be learned by scaling video generation models. This has been a topic of debate in the video generation community. For instance, OpenAI's Sora Technical Report [3] suggests that video generation models can learn rules and act as world simulators, implying they might generalize universal laws. Our findings challenge this assumption, showing that current video generation models fail to learn physical laws as universal rules.
> > 2. **Scaling guidance for Combinatorial Generalization**: We demonstrate the importance of combinatorial generalization and identify scaling laws for improving generalization in video generation models. Our findings highlight increasing the diversity of combinations in the training data is more effective than merely scaling the data volume or model size. This offers actionable guidance for building video generation models that better capture physical realism.
> > 3. We uncover how video generation models generalize, primarily relying on referencing similar training examples rather than learning underlying universal principles. This provides a deeper understanding of their limitations and biases in representing physical phenomena.
> >
> > These findings transcend traditional ID/OOD analyses, offering a more detailed understanding of the limitations and potential of video generation models. We  discuss these previous works and make our contribution clearer in appendix A.7 in our revision pdf .
> >
> > [1] Hu Y, Tang X, Yang H, et al. Case-Based or Rule-Based: How Do Transformers Do the Math?[J]. ICML 2024.

---

> > > ### Author Response · Authors · 2024-11-22
> > >
> > > `Most observations (e.g., the prioritization of color over shape) apply to general video model generalization rather than to any specific domain of physics. Consequently, the paper’s insights into generalization mechanisms are not uniquely tied to physical laws, making the connection to physical modeling somewhat tenuous.`
> > >
> > > As clarified in the first response, the primary goal of our paper is to **study the behavior of video generation models in learning physical laws from video data**. We are not aiming to develop intellectual methods for discovering general physical laws. While some physical laws can be derived solely from visual observations — for example, Newton discovered the law of universal gravitation by observing an apple fall—we specifically focus on domains where such observations are possible to train video generation models. This study can provide insightful findings, particularly in identifying the boundaries of video generation techniques, especially when applied with scaling laws.
> > >
> > >
> > > `Overall, the paper does not fully deliver on its premise of exploring physical laws, focusing instead on known generalization issues in video models without advancing the state of research in physics-informed AI.`
> > >
> > > Thank you for your feedback. We acknowledge the significant advancements in physics-informed AI. However, the primary motivation for this work stems from the success of OpenAI’s Sora, which demonstrated that video generation holds promise for building physical world models. This potential requires systematic verification.
> > >
> > > Our paper presents an empirical study to investigate whether scaling video generation models enables them to learn physical laws from raw video data, as emphasized in the abstract and introduction. **Specifically, we focus on the ability to recover physical laws purely from raw video observations, without incorporating any human-designed priors—akin to how a newborn baby learns about the world.** This approach aligns with the foundational question of whether video generation models can independently uncover universal rules of the physical world.
> > >
> > > We hope this clarifies the intent and contributions of our work. Thank you for your valuable insights!

---

> > > > ### Comment · Reviewer_r3mF · 2024-11-22
> > > >
> > > > The main weakness of this work, I think, is that none of its results are essentially new. It is a very well-known problem of deep learning models that they generalize well on ID but not well on OOD.
> > > >
> > > > My personal opinion is that for empirical study papers, (ie. papers contain only empirical studies but no new algorithms or neural arhitectures), new observations should be the main contribution. If you observed some well-known general principles in a specific domain like video generation, that is not enough to publish on a top venue like ICLR.
> > > >
> > > > I will discuss with other reviewers to see if they have any counter-arguments to this. I may revise my score accordingly, but not now.

---

> > > > > ### Author Response · Authors · 2024-11-23
> > > > >
> > > > > Thank you very much for the prompt response.
> > > > >
> > > > > The reviewer mentioned that `It is a very well-known problem of deep learning models that they generalize well on ID but not well on OOD.`
> > > > >
> > > > > We would like to emphasize that there is a distinct and crucial difference in our study. In this post-ChatGPT era, "scaling laws" are believed to be very effective at boosting the generalization abilities of models, resulting in remarkable emergent abilities in both large language models and vision foundation models like Sora. To the best of our knowledge, there is no consensus on whether **scaling** a video model will bridge the generalization gap. We are the first to investigate the generalization ability of models through scaling up models and data.
> > > > >
> > > > > Moreover, apart from the conclusions on ID and OOD settings, we also considered a special combinatorial generalization setting. Our results show that this type of generalization can benefit from scaling laws, which we believe provides insights into designing video generation models for specific downstream tasks or domains.
> > > > >
> > > > > We hope the reviewer can re-examine our paper, especially considering the aspect of "scaling."

---

> > > > > > ### Author Response · Authors · 2024-11-25
> > > > > >
> > > > > > Dear Reviewer r3mF,
> > > > > >
> > > > > > We appreciate the opportunity to address your concerns. We hope that our responses sufficiently clarify your questions, regarding the contributions of our work. With the deadline just 40 hours away, we humbly request your prompt feedback to enable further discussion.
> > > > > >
> > > > > > Best regards,
> > > > > >
> > > > > > The Authors

---

> > > > > ### Author Response · Authors · 2024-12-02
> > > > >
> > > > > Dear Reviewer r3mF,
> > > > >
> > > > > Thank you for the effort you have put into reviewing our paper.
> > > > >
> > > > > With the discussion session closing in 15 hours, we would like to take this opportunity to address your concerns and learn from your feedback.
> > > > >
> > > > > We would greatly appreciate it if you could provide comments on our rebuttals and any additional questions you may have.
> > > > >
> > > > > Best regards,
> > > > > The Authors

---

> ### Author Response · Authors · 2024-11-28
>
> Dear Reviewer r3mF,
>
> Thank you for your thoughtful comments. We would like to further address your perspective on the novelty of our paper.
> Regarding your statement, “I will discuss with other reviewers to see if they have any counter-arguments to this,” we have already had brief discussions with Reviewer 9tpf and Reviewer 47JZ, who expressed a differing view, stating: "I also read again the more critical reviews, e.g. by reviewer r3mF. I disagree with that reviewer's particular assessment of the paper..." "This paper shows several intriguing findings, improving the understanding of physical laws in video diffusion models."
>
> While we understand your concern that the relationship between ID and OOD generalization is well-known, we believe our paper makes a unique contribution by providing a deeper analysis of the nuanced behaviors of learning rules and generalization in video generation. Specifically, **our work demonstrates that generalization in video generation can be understood more subtly, particularly through our concept of "combinatorial generalization."** This form of OOD generalization performs well in many cases and exhibits scaling laws, which challenges the common belief that OOD always fails. Moreover, we offer insights into when OOD/combinatorial generalization succeeds or fails, identifying key patterns in failure cases (e.g., prioritization hierarchy in failure cases: color > size > velocity > shape). **These findings are strongly linked to the specific nature of video data, setting it apart from other forms of ID/OOD generalization.**
>
> The signals we identify regarding when and why OOD fails, and when combinatorial generalization succeeds, provides valuable perspectives for **video generation community**, given that many in the community view scaling video generation models as a way to learn accurate physical laws and expect them to generalize effectively even on unseen scenarios [1, 2].
>
> We hope this clarifies the novelty of our work, and we are happy to provide further discussion to address any additional concerns you may have. Thank you again for your valuable feedback.
>
> [1] Sora Technical Report: OpenAI. "Video Generation Models as World Simulators." Available at: https://openai.com/index/video-generation-models-as-world-simulators
>
> [2] X-AI World Model. Available at: https://www.1x.tech/discover/1x-world-model

---

### Official Review · Reviewer_9tpf · 2024-11-03

**Soundness:** 3
**Presentation:** 2
**Contribution:** 3
**Rating:** 6
**Confidence:** 4

**Summary:**

The authors study the capability of state-of-the-art video generation models (akin to SORA, combinations of VAE+DiTs) of acquiring world models from data. To this end, they create a video dataset using a basic 2D physics simulation, featuring different interacting objects. The authors study the model capability of forecasting from a given point in time, and study failure modes of video generation models. Crucially, they find that simply scaling the models is not sufficient to meaningfully improve the capability of acquiring world models, highlighting an important limitation of current video-generation techniques.

**Strengths:**

The addressed question is important, and the authors carefully design datasets and experimental settings to investigate ID/OOD behavior during model and dataset scaling, along with compositional generalisation. The overall contribution of the paper is good. The approach includes generation of a new benchmark dataset.

**Weaknesses:**

The general flow of the paper is lengthy and at times, a bit hard to parse. The overall writing needs to be improved.

I find the overall structure of the paper confusing: Sec. 2 starts with outlining some of the methods/experimental setting, then, Sec. 3 highlights ID and OOD, first introducing the setup (3.1) then the result (3.2); Sec. 4. proceeds with combinatorial generalisation, starting with the setup (4.1.) and results (4.2); Sec. 5 then mixes model setup, directly referencing results in 5.1 and each of the following sections. Given that a lot of the methodology is shared, I wonder why the authors did not write one "methods/experiments" section upfront, and then go over the different results. In particular, it would help if the headings would reflect the main results, vs. starting "main result" multiple times.

Additional points:

- I find Figure 8 hard to parse, the caption does not help much. Is this a hypothesis of what happens, or a description of the experiment?
- The paper focuses *only* on video generation models. The achieved results are not contextualised by other modeling approaches. (which might be fine within the scope of the paper, if the authors manage to contextualize the results in every figure otherwise)
- ll 74-99 already present a lot of results, without sufficient context or references. The conclusions are hard to parse. The paragraph in ll. 112-125 also seems out of place.
- The model definition in Sec 2.2 is quite slim. An overview figure would help, and more details need to be given/referenced from the appendix.
- Sec 4.2. starts with a broken sentence.

Besides, I do not see major weaknesses, I would need to clarify a few questions with the authors first to engage in further discussion, see below.

**Questions:**

- Will the dataset be fully released under a permissive license alongside the study? If you do, it would enhance adaptation a lot if all relevant latent variables from the physics simulator would be included in that release.
- What is the role of color in the different datasets? E.g. in Fig 4, you mention that color contains information about the object type (fixed vs. dynamic). What information is conveyed by color, and is this on purpose? Are there potential confounders introduced via the color?
- Sec 7: "Video generation is believed as a promising way towards scalable world models": Is this actually the case? What is scalable about video generation models? Isnt it rather the case that for video generation models, compute scales with the complexity of the data to model, and other approaches (like latent variable models, JEPA architectures, contrastive learning, ...) are the scalable models that do not need to model the input signal in full complexity?
- How was the "Abnormal rate" derived, what was the study setup? ll. 297-298 do not provide sufficient information on that.
- Larger models sometimes get larger errors (Fig 3). Could this hint at the fact that the models are undertrained? In general, could you motivate how the data sizes are picked for the different model categories, and how this relates to the amount of data these model sizes are typically trained on.
- The intro states the ordering "color > size > velocity > shape" which also takes a central role in the conclusion paragraph. Where was the "pairwise analysis" mentioned in l. 97/98 performed?

---

> ### Author Response · Authors · 2024-11-22
>
> `The general flow of the paper is lengthy and at times, a bit hard to parse. The overall writing needs to be improved.`
>
> `I find the overall structure of the paper confusing: Sec. 2 starts with outlining some of the methods/experimental setting, then, Sec. 3 highlights ID and OOD, first introducing the setup (3.1) then the result (3.2); Sec. 4. proceeds with combinatorial generalisation, starting with the setup (4.1.) and results (4.2); Sec. 5 then mixes model setup, directly referencing results in 5.1 and each of the following sections. Given that a lot of the methodology is shared, I wonder why the authors did not write one "methods/experiments" section upfront, and then go over the different results. In particular, it would help if the headings would reflect the main results, vs. starting "main result" multiple times.`
>
> Thank you for the valuable opportunity to explain the structure and flow of our paper. We would like to highlight that we are presenting an empirical study aimed at **answering whether video generation models can learn physical laws from data**. This means we are not proposing any new video generation methods; instead, we are using standard video generation models for our study. Therefore, we provide a brief description of the video generation method in the main text, referencing source papers, and leave implementation details in Appendix A.3. The main "method" for this paper is the systematic design of the study.
>
> The flow of our paper is organized into three parts:
> - (1) We define how to answer the question by examining the model's generalization ability in three scenarios: in-distribution, out-of-distribution, and combinatorial generalization (Sec. 2).
> - (2) We elaborate on the experimental settings, results, and conclusions for the three scenarios (Sec. 3 and Sec. 4).
> - (3) We provide a series of analysis experiments to reveal the generalization mechanisms of video generation models (Sec. 5).
>
> Given the exploratory nature of our work, each set of experiments often requires different settings such as task, data, and evaluation, which are strongly correlated with the results. Therefore, we present the experimental settings together with the results and analysis to ensure clarity in how each conclusion is derived. While it is possible to separate the settings and results into distinct sections, doing so may lead to confusion among readers about the different settings and conclusions.
>
> We hope our explanation helps the reviewers better understand our paper. If there are any further questions or concerns, please feel free to contact us immediately.
>
> `I find Figure 8 hard to parse, the caption does not help much. Is this a hypothesis of what happens, or a description of the experiment?`
>
> We are sorry for the confusion caused. Figure 8 is actually an illustration of the generalization results of comparing two attributes.
>
> Each subfigure compares two attributes, for example color v.s. shape. Each attribute has two distinct sets of values, for example, red and blue for color, circle and square for shape. We use red circles and blue squares for training, red squares and blue circles for testing. If a red square behaves more like a red circle, it means color is more dominating than shape in generalization. Instead, if a red square behaves more like a blue square, it means shape is more dominating than color. More experiment design details can be found in Sec. 5.3. **The arrow in Figure 8 denotes the generalization directions observed at testing time.** The bottom of Figure 8 are the corresponding examples of generated videos depicting this generalization direction.
>
> We will update the caption to make this clearer.
>
>
>
> `The paper focuses only on video generation models. The achieved results are not contextualised by other modeling approaches. (which might be fine within the scope of the paper, if the authors manage to contextualize the results in every figure otherwise)`
>
> Thank you for the kind suggestion. As the target of our study is to investigate whether video generation models can learn physical laws, and become physical world simulators. Therefore, we only focus on video generation models throughout this paper. We will make this clearer as suggested.
>
>
> `ll 74-99 already present a lot of results, without sufficient context or references. The conclusions are hard to parse. The paragraph in ll. 112-125 also seems out of place.`
>
> We'd like to clarify that line 74-99 in the introduction is a brief summary of the main results and conclusion of our paper. We will try our best to make this clearer. Meanwhile, we recommend the reviewer to go through Sec.3 ~Sec.5 for detailed experimental settings and conclusions.
>
> Line 112-125 is a formulation of the video generation problem.

---

> > ### Author Response · Authors · 2024-11-22
> >
> > `The model definition in Sec 2.2 is quite slim. An overview figure would help, and more details need to be given/referenced from the appendix.`
> >
> > We'd like to clarify that we are not proposing any new video generation methods; instead, we are using standard video generation models for our study. Therefore, we provide a brief description of the video generation method in the main text, referencing source papers, and leave implementation details in Appendix A.3.
> >
> >
> > `Will the dataset be fully released under a permissive license alongside the study?`
> >
> > Yes, we will release the datasets.
> >
> > -`What is the role of color in the different datasets? What information is conveyed by color, and is this on purpose? Are there potential confounders introduced via the color?`
> >
> > The use of color in our datasets is intentional because, in video generation, information about objects often relies on visual cues like color. In our paper, color serves distinct roles across two primary scenarios:
> >
> > 1. **Combination Generalization Experiments** (Figure 4， Section 4):  In this scenario, color encodes whether an object is dynamic or fixed. Specifically, all fixed objects are represented in black. As shown in the results, the model successfully learns this distinction.
> > 2. **Analytical Experiments on Simple Motion** ( Section 5 Deep Analysis):
> >  Here, color (red/blue) simply represents the inherent color of an object without additional meaning. We deliberately use objects with different colors in these clean, controlled setups to study what aspects video generation models rely on for generalization.
> >
> > The potential for confounders is small due to the clean design of our experimental setups. In these scenarios, color information is straightforward and easy for the diffusion model to learn.
> >
> > `Sec 7: "Video generation is believed as a promising way towards scalable world models": Is this actually the case? What is scalable about video generation models? Isnt it rather the case that for video generation models, compute scales with the complexity of the data to model, and other approaches (like latent variable models, JEPA architectures, contrastive learning, ...) are the scalable models that do not need to model the input signal in full complexity?`
> >
> > We'd like to quote the original sentence in Sora techinal report [1]: "Scaling video generation models is a promising path towards building general purpose simulators of the physical world." The remarkable success of Sora makes many people believe that scaling video generation models is promising in building world models. Therefore, we specifically focus on video generation scenarios for conducting the study.
> > We agree that there might be other ways to build world models. We leave the discussion on these works in future works.
> >
> > [1] Sora Technical Report: OpenAI. "Video Generation Models as World Simulators." Available at: https://openai.com/index/video-generation-models-as-world-simulators
> >
> >
> > `How was the "Abnormal rate" derived, what was the study setup? ll. 297-298 do not provide sufficient information on that.`
> >
> > The "abnormal rate" is determined through human evaluation, where we manually review all generated videos and count the abnormal examples. The "abnormal rate" is calculated as the number of abnormal examples divided by the total number of testing examples. For more details on the evaluation metric, please refer to Sec. 4.1 (lines 291-298).
> >
> > `Larger models sometimes get larger errors (Fig 3). Could this hint at the fact that the models are undertrained?`
> >
> > We believe this is not due to under-training, especially considering the following two observations.
> >
> > First, we compare the in-distribution testing velocity errors from Figure 3 with the VAE reconstruction errors shown in the following table. Note that we use the DiT-L model trained with 3 million videos for comparison. We observe that the in-distribution test error (e.g., 0.0124 for uniform motion) is very close to the VAE reconstruction error (e.g., 0.0105 for uniform motion). This suggests that the model is well-trained.
> >
> > |    Scenario    | VAE Reconstruction Error | In-Distribution Test Error (DiT-L/3M) |
> > |:--:|:--:|:-:|
> > | Uniform Motion |  0.0105 |   0.0124|
> > |    Collision   |0.0131 |   0.153  |
> > |    Parabola    |   0.0212 |   0.0331  |
> >
> > Second, we **plot the training loss curve of DiT-L model** for collision in Figure 16 in the revised paper. It shows the loss plateaus after 150K training steps, while our model was trained for 300K steps.

---

> > > ### Author Response · Authors · 2024-11-22
> > >
> > > `In general, could you motivate how the data sizes are picked for the different model categories, and how this relates to the amount of data these model sizes are typically trained on.`
> > >
> > > There was no artificial selection of model size or data size. For model size, we followed DiT to create a series of models with an increasing number of parameters. For data size, we generated a series of datasets with an increasing number of training examples (30K, 300K, 3M). To conduct a systematic study, we ran all combinations of datasets and model sizes, as shown in Figure 3.
> > >
> > > `The intro states the ordering "color > size > velocity > shape" which also takes a central role in the conclusion paragraph. Where was the "pairwise analysis" mentioned in l. 97/98 performed?`
> > >
> > > The pariwise analysis is conducted in Sec. 5.3 and the results are presented in Figure 8 and Figure 9.
> > >
> > > ## Summary
> > > We hope our responses have addressed your concerns regarding VAE, pretraining, and other aspects. If you have any further questions, please feel free to reach out. Thank you for your time and thoughtful review.

---

> > > > ### Author Response · Authors · 2024-11-24
> > > >
> > > > Comment:
> > > > Dear Reviewer 9tpf,
> > > >
> > > > Thank you for your constructive review on our paper. We appreciate the opportunity to address your concerns.
> > > >
> > > > We hope that our responses sufficiently clarify your questions, regarding overall writing and undertrain. With the deadline just two days away, we humbly request your prompt feedback to enable further discussion and revisions if needed.
> > > >
> > > > Best regards,
> > > >
> > > > The Authors

---

> > > > > ### Comment · Reviewer_9tpf · 2024-11-24
> > > > > **Reply to rebuttal, part 1/2**
> > > > >
> > > > > Dear authors, thanks a lot for your reply. I have some follow up questions/weaknesses to highlight:
> > > > >
> > > > > > Paper structure
> > > > >
> > > > > The current structure might be fine, after giving it a re-read.
> > > > >
> > > > > However, I would still suggest to improve the headings. Section “3.2.” and Section “4.2.” both have titles that do not tell me what I will find in the section. The style of sections in e.g. Sec 5. is much better. I would consider to improve clarity by going through the paper again, and standardizing the style of the headings. This will help readers to navigate through your paper.
> > > > >
> > > > > > Figure 8
> > > > >
> > > > > Understood — but still confusing. In particular, the *ordering* between the cartoon and the lower panel does not match. In the first columns, blue/red order is same for the graph and examples. But in column two, large/small is the order in the graph top, but in the example it is small/large, i.e. the order is flipped. In the last column, it is fast/slow in the graph, and again in the examples, first slow, then fast is shown. I would always order the examples according to the y axis used in the plot.
> > > > >
> > > > > > Re Video generation “We will make this clearer as suggested.”
> > > > >
> > > > > In which line of the updated paper is this done?
> > > > >
> > > > > > We'd like to clarify that we are not proposing any new video generation methods; instead, we are using standard video generation models for our study. Therefore, we provide a brief description of the video generation method in the main text, referencing source papers, and leave implementation details in Appendix A.3.
> > > > >
> > > > > Understood, but Appendix A.3 is equally slim. Maybe I am missing something, but from my read, there is almost no technical description of the model and its training right now. It is fine to put this in the appendix, but it is really needed. What codebase are you building on, under which license is this released? How did you train, what are the hyper parameters, optimiser, etc.? If you did not select them yourself, which paper did you adopt them from? etc. - basic information is missing here.
> > > > >
> > > > > ICLR does not provide this officially, but for guidance, it could be helpful to check the best practices in the NeurIPS paper checklist (https://neurips.cc/public/guides/PaperChecklist). It would help to add a reproducibility statement as advised in the author guide of ICLR (https://iclr.cc/Conferences/2025/AuthorGuide).
> > > > >
> > > > > > The "abnormal rate" is determined through human evaluation, where we manually review all generated videos and count the abnormal examples. The "abnormal rate" is calculated as the number of abnormal examples divided by the total number of testing examples. For more details on the evaluation metric, please refer to Sec. 4.1 (lines 291-298).
> > > > >
> > > > > Thanks. Can you please add a description of the experimental setup for measuring this metric, including the instruction sheet delivered to the human evaluators to the appendix? This is important for future replication of your metric, also also generally for judging the reported results.
> > > > >
> > > > > > Figure 16
> > > > >
> > > > > Thanks for adding. The figure is lacking a y axis label, and/or the figure caption should reference what exactly the “training loss” is in this context, which is not clear from the context. This ties into my comment above, that the experimental methods for the ML model are insufficiently described.

---

> > > > > > ### Comment · Reviewer_9tpf · 2024-11-24
> > > > > > **Reply to rebuttal, part 2/2**
> > > > > >
> > > > > > > The pariwise analysis is conducted in Sec. 5.3 and the results are presented in Figure 8 and Figure 9.
> > > > > >
> > > > > > Thanks, especially with the description in the rebuttal, I better understand the rationale. However, I find this still not optimally presented.
> > > > > >
> > > > > > Let me recap:
> > > > > >
> > > > > > - In Fig. 8, the experimental design creates a conflict between color and shape. The two possible, equally likely experiment outcomes would be: The blue ball could convert in a (1) red ball or (2) blue square, depending on which feature is “stronger”. ALTERNATIVELY, an optimal solution would be that they remain blue balls. With this setup, I see:
> > > > > >     - (1) color > shape
> > > > > >     - (2) size > shape
> > > > > >     - (3) velocity > shape
> > > > > >     - And can conclude that {color, size, velocity} > shape, which you note in ll 409-418. **This part is fine!** (although it would be much better to do this quantitatively)
> > > > > > - In Fig. 9,
> > > > > >     - The velocity of some, very extreme frames changes while the radius is relatively constant. From this you conclude, size > velocity.
> > > > > >     - Now we compare size vs. color. Color is stable, but the radius seems stable as well? It seems like there is no switch from large to small. Do you have example frames? A better experiment here would be to pick two sizes (large and small) like in Figure 8, and then make a “binary” comparison how many balls are assigned to a different size. Your conclusion here still seems “color > size”, although this is very slim.
> > > > > >     - The last panel shows a drift, but still, there are no points really moving into the regime of the other attribute, i.e., no “blue, small velocity” samples are truly converted into “blue, high velocity”. Your conclusion is still, “color > velocity”
> > > > > >
> > > > > > Overall, I hope I correctly inferred your rationale for saying “color > size > velocity > shape”.
> > > > > >
> > > > > > If my reasoning matches yours, I do not find this backed up in the strong form you make in the paper.
> > > > > >
> > > > > > 1. The paradigm is fundamentally ill-posed and set up such that “binary attributes” like color are more likely to be dominant. It is a big confounder that some attributes are binary (color, shape) and some are continuous (size, velocity), which is not discussed in the paper (correct me if wrong).
> > > > > > 2. In all examples of Figure 9, only the “border cases” show a meaningful drift. How do you currently confirm that this is indeed due to the cue conflict, and not general behavior? What is the control experiment? I.e., if you repeat the experiments in panel (2) and (3) without a cue conflict (i.e. just red or just blue), how does the curve look like? Is there still a tendency that the extreme cases move to the center of the distribution?
> > > > > > 3. Follow up Q: This whole analysis only makes sense if there is a true cue conflict in the dataset setup. Apologies if I overread this, but can you confirm that for each train dataset used in Figure 8, 9, the training data contains a valid continuation for both hypotheses?
> > > > > > 4. I would strongly advise to include a quantitative analysis, which should be still doable before the end of the rebuttal period as it does not require any retraining.
> > > > > >
> > > > > > I hope this gives some direction for further improvement. Overall, I see the option to (1) conduct new analysis, or (2) tune down the statements in the paper about the conclusion, and add a limitation section outline all the possible confounders.
> > > > > >
> > > > > > Happy to discuss further.

---

> > > > > > > ### Author Response · Authors · 2024-11-25
> > > > > > >
> > > > > > > Thank you for your time and prompt response. See the following for your questions:
> > > > > > >
> > > > > > > `Headings for section 3.2 and section 4.2`
> > > > > > >
> > > > > > > Thank you for the valuable suggestion. The headings for section 3.2 is renamed to "Perfect ID and Failed OOD Generalization", while the heading for section 4.2 is renamed to "Scaling Law Observed for Combinatorial Generalization"
> > > > > > >
> > > > > > > `In Figure 8,  the ordering between the cartoon and the lower panel does not match`
> > > > > > >
> > > > > > > Thank you for the suggsions. The figure is updated accordingly. Please see our revised draft.
> > > > > > >
> > > > > > > `Re video generation "We will make this clearer as suggested."`
> > > > > > >
> > > > > > > We emphasized our focus on studying the problem of whether video generation models can learn physical laws in both the abstract (lines 010-012) and the introduction (lines 041-043). **Additionally, we reiterate this at the beginning of each section, highlighted in blue (lines 128-130, 170, 256-257).** Please refer to our revised version. Regarding the addition of this description for each figure, we seek your advice on whether this might be redundant given the numerous figures in our paper.
> > > > > > >
> > > > > > > `Re appendix A.3 is quite slim`
> > > > > > >
> > > > > > > At the beginning of section 2.2, we emphasized that we are following Sora to use a VAE model and DiT architectures for video generation. The detailed configs DiT are given in Table 1 in the main text. **According to your suggestions, we added training recipe in Appendix A.3.3.**
> > > > > > >
> > > > > > > `Setup and instructions for human evolution for "abnormal rate"`
> > > > > > >
> > > > > > > Thanks for your advice. We have added setup and instructions for human evaluation for "abnormal rate" in Appendix A.4.2.
> > > > > > >
> > > > > > > `Figure 16 is lacking a y axis label, and/or the figure caption should reference what exactly the “training loss” is in this context`
> > > > > > >
> > > > > > > The training loss is defined in Appendix A.2 with Equation (2). We have updated the figure in the appendix and revised the caption to reference this definition.
> > > > > > >
> > > > > > > ---
> > > > > > > ## Improvement for Pariwise analysis of the generalization order
> > > > > > >
> > > > > > > `Your conclusion here still seems “color > size”, although this is very slim,  A better experiment here would be to pick two sizes (large and small) like in Figure 8, and then make a “binary” comparison;
> > > > > > >  ill-posed set up that some attributes are binary (color, shape) and some are continuous (size, velocity)`
> > > > > > >
> > > > > > > Thanks for your constructive suggestion. We acknowledge this point and are conducting additional experiments where we select two discrete sizes or velocities to create binary comparisons. Once these experiments conclude, we will include the results in the paper and share them with you.
> > > > > > >
> > > > > > > `How do you currently confirm that this is indeed due to the cue conflict, and not general behavior? What is the control experiment?`
> > > > > > >
> > > > > > > Thanks for your suggestion. We acutally have run  the experiments without a cue conflict (i.e. just red). **These results are now added in the revised manuscript under Appendix A.6.2 ("Control Experiment for Pairwise Comparison") and illustrated in Figure 14 in revision**. The experiments reveal that even in extreme cases, predictions remain accurate without any shift. This supports our conclusion that the drift in Figure 9 is due to cue conflict and the presence of attribute prioritization.
> > > > > > >
> > > > > > > `can you confirm that for each train dataset used in Figure 8, 9, the training data contains a valid continuation for both hypotheses?`
> > > > > > >
> > > > > > > We carefully ensured that all datasets in Figures 8 and 9 include true cue conflicts in the training data. For example, in the "color vs. shape" case, the training set only contains (1) red balls and (2) blue squares, with an equal number of training examples for both. Thus, during inference, a blue ball could logically transform into either (1) a red ball or (2) a blue square, depending on the stronger feature. This setup guarantees a valid cue conflict for testing attribute prioritization.
> > > > > > >
> > > > > > > `I would strongly advise to include a quantitative analysis`
> > > > > > >
> > > > > > > Thank you for the suggestion. If we understand correctly, the quantitative analysis you refer to pertains to Figure 8, where we conclude that {color, size, velocity} > shape. Since shape and color are binary attributes, we quantified the results by providing the ratio of transformations into each binary class as a measure.
> > > > > > >
> > > > > > > We observed no exceptions across 1400/620/704 test cases where the shape transformed visibly, while color, size, or velocity remained consistent, as stated in line 414 of the paper. Color consistency is visually apparent, while size and velocity were parsed from the generated videos and compared against the conditions, showing negligible changes.
> > > > > > >
> > > > > > > If you have further suggestions for additional quantitative analyses, we would be happy to consider and incorporate them.
> > > > > > >
> > > > > > > ---
> > > > > > >  I hope these points help address your questions.

---

> > > > > > > > ### Comment · Reviewer_9tpf · 2024-11-25
> > > > > > > >
> > > > > > > > Thanks for the swift reply. Follow up:
> > > > > > > >
> > > > > > > > > A.3.3
> > > > > > > >
> > > > > > > > Thanks for adding, much better. For the purpose of reproducibility, did you build on an open source implementation? If so, from where?
> > > > > > > >
> > > > > > > > Will the training and evaluation code associated to the paper made fully open source upon publication?
> > > > > > > >
> > > > > > > > > l 918 “Horizontal flipping is the only data augmentation if not specified.”
> > > > > > > >
> > > > > > > > Isn’t this a huge confounder given that you have horizontal movement in the video? What is the rationale? Is this done on the level of the whole video, or single images
> > > > > > > >
> > > > > > > > > A.4.2
> > > > > > > >
> > > > > > > > Thanks for adding this — this adds a lot of important context.
> > > > > > > >
> > > > > > > > One follow up, were human evaluators given the opportunity to watch the video multiple times/as they wished, or did they only have a single shot watching the video?
> > > > > > > >
> > > > > > > > Could they e.g. pause the video, or was it played at “real time” speed? How long was each video in terms of seconds?
> > > > > > > >
> > > > > > > > Will you share the raw ratings of this evaluation alongside the dataset?
> > > > > > > >
> > > > > > > > > I would strongly advise to include a quantitative analysis
> > > > > > > >
> > > > > > > > I think in particular for figure 9, this would be interesting, figure 8 and the text in the paper is sufficiently convincing. Maybe it is something that can be tackled once the binary condition experiments are run.
> > > > > > > >
> > > > > > > > Looking for the experiment results.
> > > > > > > >
> > > > > > > > Paper is already much stronger with the additions. I will give you a firm update on my final score before the official end of the discussion period, but will wait for any further change you make based on the experiments currently running before that.
> > > > > > > >
> > > > > > > >
> > > > > > > > Minor
> > > > > > > >
> > > > > > > > > l 1077: not due to …
> > > > > > > > > l 1078: typo, “an attribute …”

---

> > > > > > > > > ### Author Response · Authors · 2024-11-25
> > > > > > > > >
> > > > > > > > > Thank you for your time and prompt response. I hope the following points help address your questions.
> > > > > > > > >
> > > > > > > > > `A.3.3 For the purpose of reproducibility, did you build on an open source implementation? If so, from where?`
> > > > > > > > >
> > > > > > > > > Our code is derived from an internal codebase, which references the DiT repository ( https://github.com/facebookresearch/DiT). However, we regret that the internal codebase cannot be submitted for review in this stage due to internal policy. Upon publication, we will clean and open source the training and evaluation scripts.
> > > > > > > > >
> > > > > > > > > `l 918 “Horizontal flipping is the only data augmentation if not specified.” Isn’t this a huge confounder given that you have horizontal movement in the video? What is the rationale? Is this done on the level of the whole video, or single images`
> > > > > > > > >
> > > > > > > > > The rationale behind horizontal flipping lies in the spatial invariance of physical laws. For example, in the case of uniform motion, flipping generates a symmetrical physical event that still adheres to the principles of physics, thereby augmenting the training data effectively. This augmentation is applied at the level of the whole video, ensuring that the flipped version represents a spatially symmetrical physical event.
> > > > > > > > >
> > > > > > > > > `A.4.2 were human evaluators given the opportunity to watch the video multiple times/as they wished, or did they only have a single shot watching the video? Could they e.g. pause the video, or was it played at “real time” speed? How long was each video in terms of seconds? Will you share the raw ratings of this evaluation alongside the dataset?`
> > > > > > > > >
> > > > > > > > > Yes, human evaluators were allowed to watch the videos multiple times and pause them as they wished. However, the instruction sheet encouraged them to make judgments based on their first impression to reflect intuitive physics. The raw ratings will be released later upon the necessary approval.
> > > > > > > > >
> > > > > > > > > `I would strongly advise to include a quantitative analysis`
> > > > > > > > >
> > > > > > > > >  We are trying our best to implement and conduct the binary condition experiments.
> > > > > > > > >
> > > > > > > > > `Minor：l 1077: not due to … l 1078: typo, “an attribute …”`
> > > > > > > > >
> > > > > > > > > Thank you for pointing these out. We have corrected them in the revision.

---

> > > > > > > > > > ### Author Response · Authors · 2024-11-26
> > > > > > > > > >
> > > > > > > > > > ## Binary Condition Experiments
> > > > > > > > > > Based on your constructive suggestion, we’ve restructured the experiments to incorporate binary comparisons, which should offer a clearer analysis of the prioritizations between the attributes.
> > > > > > > > > >
> > > > > > > > > > ### Color vs. Velocity
> > > > > > > > > > In the training set, we have: (1) Red, slow ball (2) Blue, fast ball
> > > > > > > > > >
> > > > > > > > > > During evaluation, we observe the following transformations: (1) All **red, fast** balls are transformed into **red, slow** balls. (2) All **blue, slow** balls are transformed into **blue, fast** balls. We conducted 202 evaluation cases in total, and in each case, the velocity changes while the color remains unchanged. This supports the conclusion that color > velocity since the transformation is fully governed by velocity without affecting the color attribute.
> > > > > > > > > >
> > > > > > > > > > ### Size vs. Velocity
> > > > > > > > > > In the training set, we have: (1) Small, slow ball (2) Large, fast ball
> > > > > > > > > >
> > > > > > > > > > During evaluation: (1) All **small, fast** balls are transformed into **small, slow** balls. (2) All **large, slow** balls are transformed into **large, fast** balls. Again, we observed 202 evaluation cases, where velocity was transformed while size remained fixed. This clearly shows that size > velocity, as the transformation is governed by velocity while size stays constant.
> > > > > > > > > >
> > > > > > > > > > ### Color vs. Size
> > > > > > > > > > We are still in the process of running this experiment, but we will update the results once the analysis is complete.
> > > > > > > > > >
> > > > > > > > > > We hope these binary comparisons help clarify our conclusions. Once all the experiments are complete, we will include the results in the revised version. Thank you again for your valuable input!

---

> > > > > > > > > > > ### Comment · Reviewer_9tpf · 2024-11-26
> > > > > > > > > > >
> > > > > > > > > > > Thanks, that sounds quite exciting and def supports the story. I’ll stay put for the final revised version then.

---

> > > > > > > > > > > > ### Author Response · Authors · 2024-11-27
> > > > > > > > > > > >
> > > > > > > > > > > > Dear reviewer 9tpf,
> > > > > > > > > > > >
> > > > > > > > > > > > We have completed the binary experiment for color vs. size. In the training set, we used: (1) red, small balls and (2) blue, large balls. During evaluation: (1) all **red, large** balls were transformed into **red, small** balls, and (2) all **blue, small** balls were transformed into **blue, large** balls. We observed 202 evaluation cases, where size was transformed while color remained fixed, clearly demonstrating that color > size.
> > > > > > > > > > > >
> > > > > > > > > > > > Additionally, we have updated the revision, and these binary experiments can be found in **Section 5.3 and Figure 9**. We hope these binary comparisons help clarify our conclusions and strengthen your evaluation of our paper. Thank you once again for your time and valuable suggestions.

---

> ### Comment · Reviewer_9tpf · 2024-11-27
>
> Dear authors, thanks a lot for adding. I increased the scores for soundness and presentation, as well as my overall rating of the paper based on the conducted changes. I think that the original claims of the paper are much more convincingly presented now, and also the additions to the experimental setup are useful.
>
> ---
>
> I also read again the more critical reviews, e.g. by reviewer r3mF. While I disagree with that reviewer's particular assessment of the paper, I wanted to touch on this part of your reply,
>
> > We would like to emphasize that there is a distinct and crucial difference in our study. In this post-ChatGPT era, "scaling laws" are believed to be very effective at boosting the generalization abilities of models, resulting in remarkable emergent abilities in both large language models and vision foundation models like Sora. To the best of our knowledge, there is no consensus on whether scaling a video model will bridge the generalization gap. We are the first to investigate the generalization ability of models through scaling up models and data.
>
> > Moreover, apart from the conclusions on ID and OOD settings, we also considered a special combinatorial generalization setting. Our results show that this type of generalization can benefit from scaling laws, which we believe provides insights into designing video generation models for specific downstream tasks or domains.
>
> as well as this sentence from the abstract:
>
> > Our study suggests that scaling alone is insufficient for video generation models to uncover
> fundamental physical laws, despite its role in Sora’s broader success.
>
> I think that these statements are too strong. Doesn't your result on combinatorial generalisation indeed show that scaling helps, in these cases? I think the sentence in the abstract does not fully align with the results you actually show.

---

> ### Author Response · Authors · 2024-11-27
>
> Thank you very much for reconsidering your ratings and for supporting the acceptance of our paper.
>
> Regarding your concern about the strength of our statements, could you please clarify which specific sentences you find to be too strong in the first two points? We would greatly appreciate your input on this.
>
> As for the third point regarding the abstract, thank you for highlighting this issue.  We'd like to provide some context for why we structured it this way. This actually depends on how we define if a physical law is discovered or not. Typically, a universal physical law should be applicable to any situation (ID, OOD, and combinatorial settings). However, video generation models apparently cannot achieve this. We agree that the original statement in the abstract may look a bit strong without context.  Therefore, we propose revising the statement to explicitly point out the requirements for being considered universal physical laws. What do you think of this approach?
>
> > Our study suggests that scaling alone is insufficient for video generation models to uncover universal physical laws applicable to all scenarios (or "to OOD scenarios")
>
> Thank you again for your valuable feedback, and we look forward to hearing your thoughts on these revisions.

---

> ### Author Response · Authors · 2024-12-02
>
> Dear Reviewer 9tpf,
>
> Thank you for the effort you have put into reviewing our paper.
>
> With the discussion session closing in 15 hours, we would like to take this opportunity to address your concerns and learn from your feedback.
>
> We would greatly appreciate it if you could provide comments on our rebuttals and any additional questions you may have.
>
> Best regards,
> The Authors

---

### Official Review · Reviewer_ym7u · 2024-11-03

**Soundness:** 1
**Presentation:** 3
**Contribution:** 2
**Rating:** 5
**Confidence:** 4

**Summary:**

The authors train their own video diffusion models on synthetic video models and run thorough analysis on how the physics based video generation.

**Strengths:**

* The paper train many video diffusion models from scratch on small physical datasets across model parameter sizes.
* They have thorough evaluation of extrapolation behavior
* The paper attempts to tackle a very complex problem, by simplifying it to a synthetic data setting.
* The analysis between in distribution and OOD seems useful, and this distinction could inform the creation of future video models, particularly the experiments in 5.4

**Weaknesses:**

* Instead of training a "small" video model from scratch, why not try finetuning SOTA models on these video datasets? One issue with this analysis it supposes that there is not a minimum threshold for the number of parameters needed for a useful diffusion video model or for generalization to hold. I would not be surprised if these models generalized when simply having more parameters and train on more data, even out of domain data. Finetuning a model like SVD should be doable on a similar level of compute.

* The rendering of the synthetic examples are overly simplistic and do not have imaging artifacts that video models could exploit in real world use cases to generalize, like motion blur.

* Many of these reasoning weakness of generative models have been brought before in other domains. Such as Arc-AGE Challenge - "On the Measure of Intelligence" by François Chollet.

*  "For example, in Figure 10, it is difficult to determine if a ball can pass through a gap based on vision alone when the size difference
is at the pixel level, leading to visually plausible but incorrect results. Similarly, visual ambiguity in a ball’s horizontal position relative to a block can result in different outcomes. These findings suggest that relying solely on visual representations, may be inadequate for accurate physics modeling." Pixel level differences will be obliterated by the VAE encoder. It is a compressive architecture, information will be lost to train the model more cheaply. Pixel level criteria is not a motivating example as a result. If you want pixel level accuracy, you need pixel level diffusion models, not one trained on latent. I would disregard these experiments or rewrite this entire section 5.5 as this is a fundamental issue with the model architecture, and reveals no new information. Can you at least verify that the VAE is able to reconstruct different images with pixel level differences? I think you will find that it will not, as Stable Diffusion's VAE architecture cannot. The number of channels is way too limiting which is why it's been massively increased in more recent models like BlackForrest's FLUX.

* " This ranking could explain why current video generation models often struggle with maintaining object consistency" Please support with evidence? Or at least specify with which video models?


* Our in-depth analysis suggests that video model generalization relies more on referencing similar training examples rather than learning universal rules.
All generative models exhibit this behavior, and it is well known. Similar observations can be made on small language models, but they usually generalize way better when scaled up in terms of compute. What is the new observation here with respect to physics?


Rather a better way to frame this paper might be what role do VAE reconstruction issues prevent us from using existing video world models for physics in critical settings? How do the video VAE's and the diffusion model prioritize shape color and velocity? The main issue here is that some of the experiments are clearly demonstrating architectural failures of the VAEs and the authors are attempting to generalize it to all video models, which is an massive overclaim.

This paper has useful experimental and scientific data, but it needs to be rewritten to support the claims in the paper. Furthermore, it could massively benefit by examining which of these issues are coming from the VAE and which are coming from the latent video diffusion.

Claims are made that scaling cannot solve these physics problem. The evidence this paper shows that scaling parameters and perhaps date in the video diffusion model is ineffective if the VAE removes important information for the video model.

**Questions:**

* In what real world cases would the data be significantly outside of the distribution of the training data. Should the solution not be scaling the number of parameters or the number of data, but rather the diversity of the data then?

* What aspects of the behavior you observation are purely artifacts of the latent encoder / decoder?

* Why did you not finetune any existing video models on these tasks? Surely this task is entirely OOD from the training task since it's a synthetic environment. Furthermore, if as a researcher your goal is to train the best video world model possible, why would you not start with a pretrained model. If you are confident that these problems cannot be solved purely by scaling, than pretrained video models on real world images should not generalize to this simple benchmark and should observe similar biases as in this paper. If you are making a claim that this behavior can be observed on all video models, than you should be able to evaluate this lack of generalizations on existing pre-trained models right?


* We design datasets which delibrately leave out some latent values, i.e. velocity. After training, we test model’s prediction on both seen and unseen scenarios. We mainly focus on uniform motion and collision processes
From an optimization point of view, what sparsity level is needed for the model to be able to linearly extrapolate then? After all, a diffusion model must also know what NOT to generate. There is surely some level sparsity for which the video model will generalize? What level of sparsity is it?

* "color > size > velocity > shape" how much of this is simply affected by this explicit instantiation of the video VAE? What about other ones like Stable Video Diffusions? Or more recently released video model like COGVideo's?

---

> ### Author Response · Authors · 2024-11-22
>
> Thank you for your insightful comments. We appreciate the opportunity to address your concerns regarding scaling, generalization, and VAE in our paper.
>
> `Q1: Why train a video gen model from scratch instead of finetuning pretrained SOTA models? How would the findings in the paper apply to finetuning open-source pretrained models, such as Stable Video Diffusion (SVD)?`
>
> - Please refer to the global response for a detailed discussion on this question.
>
> `Q2: One issue with this analysis it supposes that there is not a minimum threshold for the number of parameters needed for a useful diffusion video model or for generalization to hold. I would not be surprised if these models generalized when simply having more parameters and train on more data, even out of domain data.`  `If you are confident that these problems cannot be solved purely by scaling, than pretrained video models on real world images should not generalize to this simple benchmark and should observe similar biases as in this paper. If you are making a claim that this behavior can be observed on all video models, than you should be able to evaluate this lack of generalizations on existing pre-trained models right? `
>
> 1. **Low Likelihood of Emergence in Our Setting**: We agree that emergent abilities in models can arise under certain conditions, as explained in [1]. According to this reference, emergent abilities typically result not from fundamental changes in model behavior with scale but from the researcher’s choice of metrics. Specifically, **nonlinear or discontinuous metrics** tend to produce the illusion of emergent abilities, while linear or continuous metrics yield smooth, predictable performance changes with scale.
>    In our study, we took care to use metrics that avoid such artifacts. **Our velocity error metric is linear and continuous**, as it averages across frames, ensuring that the observed results reflect genuine model behavior rather than artifacts of the evaluation metric. In ID/OOD experiments, we found no evidence of consistent or significant OOD error reduction as model size or training data increased. This empirical observation supports our conclusion that simply scaling model parameters or datasets is insufficient to address the lack of generalization observed in our experiments.
>
>
> 2. **Additional Experiments**:To further substantiate our claim, we conducted additional experiments to investigate the effects of model scaling and fine-tuning:
> - **Scaling Models to DiT-XL**: We extended our analysis to include a larger DiT-XL model trained on the 3M  uniform motion video dataset. While the in-distribution error decreased with scale, the OOD error showed no consistent improvement and, in some cases, increased slightly.
> - **Fine-Tuning Stable Video Diffusion (SVD)**: We fine-tuned the pretrained SVD models on our 3M dataset and evaluated it. As shown in the table below, even with pretraining, the lack of generalization in OOD settings persisted, further highlighting the limitations of scaling in addressing this issue.
>
>
> | Model                 | In-Distribution Error | OOD Error |
> |-----------------------|------------------------|-----------|
> | DIT-S (3M)           | 0.0149                | 0.2875    |
> | DIT-B (3M)           | 0.0138                | 0.3583    |
> | DIT-L (3M)           | 0.0124                | 0.4270    |
> | DIT-XL (3M)          | 0.0119                | 0.3952    |
> | SVD Fine-tune (3M)   | 0.0505                | 0.9081    |
> | Ground Truth         | 0.0099                | 0.0104    |

---

> > ### Author Response · Authors · 2024-11-22
> >
> > `Q3: it could massively benefit by examining which of these issues are coming from the VAE and which are coming from the latent video diffusion. `
> >
> > **VAE’s Ability to Preserve Important Kinematic Information**:
> > We appreciate the importance of verifying that the VAE does not significantly contribute to the issues observed in our paper. As detailed in Appendix A.3.2 and summarized here, our experiments confirm that the pretrained VAE encoder accurately preserves essential kinematic information. Specifically, we evaluated the VAE’s reconstruction performance by encoding and decoding ground truth videos and using the reconstructed videos to parse velocity. We then calculated the error between the parsed velocity and the values from the simulator’s internal state.
> >
> > The results show that the reconstruction error ($e_{\text{recon}}$) is very close to the ground truth error ($e_{\text{gt}}$), where $e_{\text{gt}}$ represents the error between parsed velocity from the ground truth video and the simulator values. Both errors are an order of magnitude lower than the OOD error observed in the DiT-B model. These results demonstrate that the VAE preserves key kinematic information with high fidelity, and the issues discussed in our paper are attributable to the latent video diffusion model rather than the VAE itself.
> >
> > | Scenario        | GroundTruth Error ($e_{\text{gt}}$) | VAE Recon Error ($e_{\text{recon}}$) | OOD Error (DiT-B) |
> > |-----------------|---------|--------------|-------------------|
> > | Uniform Motion  | 0.0099   | 0.0105        | 0.3583            |
> > | Collision       | 0.0117      | 0.0131           | 0.2106            |
> > | Parabola        | 0.0210      | 0.0212        | 0.2881            |
> >
> >
> >
> > `Q4: Can you at least verify that the VAE is able to reconstruct different images with pixel level differences? I think you will find that it will not, as Stable Diffusion's VAE architecture cannot. The number of channels is way too limiting which is why it's been massively increased in more recent models like BlackForrest's FLUX.`
> >
> > Thank you for your valuable suggestions. We conducted an experiment to verify whether the VAE used in our study can reconstruct images with pixel-level differences. Specifically, we generated images where the position of a ball differed by a single pixel and evaluated the reconstructed images to determine if the VAE preserved these differences.  The ball’s center position was calculated as the average of the coordinates of the colored pixels in the ball as we did in the paper. We tested three pairs of images to ensure reliability and observed the following results:
> >
> > | Ground Truth Ball Center (Pixel) | Reconstructed Ball Center (Pixel) |
> > |--|---|
> > | 50   | 50.043  |
> > | 50 | 50.056   |
> > | 50    | 50.034  |
> > | 51    | 51.014 |
> > | 51  | 50.986   |
> > | 51   | 50.987    |
> >
> > These results demonstrate that the VAE effectively preserves pixel-level differences in ball positions. While slight deviations exist, they are minor and do not obscure the pixel-level distinctions between the input images.  We attribute this capability to the simplified visual setting in our experiments, which deliberately removes complex textures and contours, leaving only pure-colored balls on a white background. This controlled design reduces unnecessary information, enabling the VAE to focus on retaining a minimal amount of essential information.
> >
> > We will include these findings in the revised version of the paper. Thank you again for pointing out this area  to improve the robustness of our approach.
> >
> > `Q5: The rendering of the synthetic examples are overly simplistic and do not have imaging artifacts that video models could exploit in real world use cases to generalize, like motion blur.`
> >
> > Thank you for your thoughtful feedback. As you pointed out, motion blur can indeed provide valuable clues about speed and direction, which models may exploit to generalize in real-world use cases. However, in our setup, the first 4 frames are provided as conditions, enabling the model to infer these dynamics directly. Additionally, using simplified synthetic examples offers the following advantages:
> >
> > 1. **Accurate VAE Modeling**: As demonstrated in our responses above, the simplified examples allow the VAE to accurately preserve motion information. This ensures that our analysis can focus on how the diffusion model learns physical laws without the confounding effects of VAE reconstruction errors.
> >
> > 2. **Abundance of Controllable Data**: Synthetic examples provide a large volume of highly controllable data, enabling systematic exploration of specific physical principles and behaviors under well-defined and reproducible conditions.
> >
> > By simplifying the rendering process, we isolate and address the core challenges of learning physical laws, making our findings more interpretable and focused. We appreciate your valuable insights and welcome further discussion.

---

> > > ### Author Response · Authors · 2024-11-22
> > >
> > > `Q6: "color > size > velocity > shape" how much of this is simply affected by this explicit instantiation of the video VAE? What about other ones like Stable Video Diffusions? Or more recently released video model like COGVideo's?`
> > >
> > > Thank you for your insightful feedback. To address the concern, we have conducted additional experiments using the VAE from CoGVideo. Specifically, we replaced the VAE in our original setup with CoGVideo's VAE, froze its parameters, and trained the diffusion model.
> > >
> > > Due to the time constraints of the rebuttal phase, we focused on three key comparisons: color vs. size, size vs. velocity, and velocity vs. shape, instead of the six experiments in our original paper. Despite the architectural and pretrain differences, the prioritization order observed in our paper (color > size > velocity > shape) remained consistent. The detailed experimental results are provided in Appendix A.5.1 of our updated paper PDF.
> > >
> > >  If you have further concerns about the influence of the VAE on our results or any other aspects, please do not hesitate to reach out. We appreciate your valuable feedback.
> > >
> > > `Q7: " This ranking could explain why current video generation models often struggle with maintaining object consistency" Please support with evidence? Or at least specify with which video models?`
> > >
> > > Thank you for your feedback. Below, we provide evidence to object consistency challenges:
> > >
> > > - **Sora Technical Report** ([3]): Reports "We enumerate other common failure modes of the model—such as incoherencies that develop in long duration samples or spontaneous appearances of objects"
> > > - **X-AI World Model** ([4]): Notes failures in "maintaining the shape of objects during interaction," including object disappearance and distortion under occlusion or unfavorable angles.
> > > - **ConsistI2V** ([5]): Highlights "existing methods often struggle to preserve the integrity of the subject, background, and style from the first frame"
> > >
> > > [3] Sora Technical Report: OpenAI. "Video Generation Models as World Simulators." Available at: https://openai.com/index/video-generation-models-as-world-simulators
> > > [4] X-AI World Model. Available at: https://www.1x.tech/discover/1x-world-model
> > > [5] Ren, W., Yang, H., Zhang, G., Wei, C., Du, X., Huang, W., & Chen, W. "ConsistI2V: Enhancing Visual Consistency for Image-to-Video Generation." Transactions on Machine Learning Research 2024.
> > >
> > > `Q8: Our in-depth analysis suggests that video model generalization relies more on referencing similar training examples rather than learning universal rules. All generative models exhibit this behavior, and it is well known. Similar observations can be made on small language models, but they usually generalize way better when scaled up in terms of compute. What is the new observation here with respect to physics?`
> > >
> > > Thank you for your feedback. We respectfully disagree with the statement "All generative models exhibit this behavior (generalization relies more on referencing similar training examples rather than learning universal rules)" as a general conclusion. **Specifically for video generation, there has not been a clear consensus in prior works**. For example, OpenAI's Sora Technical Report [3] claims that video generation models can learn rules and act as world simulators, suggesting they might be capable of learning universal rules.
> > >
> > > **One of the key contributions of our paper is to explicitly demonstrate that video generation models do not learn physical laws as universal rules via systematical experiment designs.** Through analytical experiments, we further reveal what these models are actually learning—primarily case-based memorization. In addition, we analyze how video generation models prioritize certain attributes to reproduce this memorization.
> > >
> > > **Another significant contribution of our work is uncovering the scaling laws for combinatorial generalization in video generation models.** We show that scaling the diversity of training data can help generate physics-consistent videos in combinatorial scenarios. This provides actionable guidance for improving video generation models by increasing data diversity to enhance their ability to simulate physical processes.
> > >
> > > These findings offer new insights into the limitations and potential of video generation models in capturing and reproducing physical realism. Thank you again for your thoughtful comment!

---

> > > > ### Author Response · Authors · 2024-11-22
> > > >
> > > > `Q9: Many of these reasoning weakness of generative models have been brought before in other domains`
> > > >
> > > > We acknowledge that reasoning weaknesses in machine learning have been extensively discussed in foundational works. These discussions highlight the fundamental challenges of reasoning in machine learning.
> > > >
> > > > **However, when applied to specific settings with unique data forms, distributions, and model inductive biases, reasoning capabilities and failure modes exhibit nuanced behaviors that remain scientifically significant.** For instance, similar debates have emerged in the study of large language models (LLMs) to determining whether models learn underlying rules or rely on case-based memorization [6]. These insights have spurred advancements in rule-based reasoning approaches within the LLM community. Similarly, our work investigates how video generation models handle generalization.
> > > >
> > > > Our findings go beyond the simplistic observation that ID scenarios succeed while OOD scenarios fail. Instead, we provide deeper insights and discoveries specific to video generation:
> > > >
> > > > - **Scaling in Combinatorial Generalization**: We investigate the models' ability to generalize across combinatorial scenarios and demonstrate that increasing the diversity of combinations in training data can significantly enhance performance.
> > > > - **Inductive Biases and Failure Modes:** We analyze the specific biases inherent in video generation models and provide detailed insights into failure cases, such as their difficulty in preserving object shapes during dynamic interactions.
> > > >
> > > > These findings transcend traditional ID/OOD analyses, offering a more detailed understanding of the limitations and potential of video generation models.
> > > >
> > > > [6] Hu Y, Tang X, Yang H, et al. Case-Based or Rule-Based: How Do Transformers Do the Math?[J]. ICML 2024.
> > > >
> > > > `Q10: In what real world cases would the data be significantly outside of the distribution of the training data. Should the solution not be scaling the number of parameters or the number of data, but rather the diversity of the data then?`
> > > >
> > > > Our work focuses on scientific findings, making it crucial to consider the underlying mechanisms in various scenarios. Specifically, when using video generation models as world simulators, real-world cases often involve data significantly out of distribution (OOD). Examples include:
> > > >
> > > > 1. **Robotics Data**: Robots frequently encounter scenarios involving varying object velocities, joint angles, and complex interactions with objects. These attributes can vary significantly, and the vast combinations of scenario elements create diverse environments, often leading to situations far outside the training data distribution.
> > > > 2. **Autonomous Driving**: Corner cases, such as unusual weather conditions, rare object appearances, and unpredictable interactions, often vary significantly in severity and consistently fall outside the training data distribution.
> > > >
> > > > In these contexts, our paper’s results suggest the following insights:
> > > > - **combination Diversity**: Once the dataset reaches a sufficient scale, increasing its diversity is more impactful than merely scaling the volume of similar data. Diversity allows the model to handle unseen combinatorial cases.
> > > > - **Scaling Model Parameters**: Increasing model parameters proves beneficial, if the data is diverse enough to supply meaningful information to learn from.
> > > >
> > > > `Q11: From an optimization point of view, what sparsity level is needed for the model to be able to linearly extrapolate then? After all, a diffusion model must also know what NOT to generate. There is surely some level sparsity for which the video model will generalize? What level of sparsity is it?`
> > > >
> > > > Thank you for raising this intriguing point about the sparsity level required for effective generalization and interpolation. As illustrated in Figure 5 (1) and (2) of our paper, when the sparsity gap constitutes 5/6 and 2/3 of the training range, the model struggles to interpolate accurately. However, when the gap is reduced to 1/2 of the training range (as shown in Figure 5 (3), where the training range is [1, 4] and the gap is 1.5), the model performs significantly better, even with a relatively large gap.
> > > >
> > > > Additionally, as demonstrated in Figure 5 (4) and (5), incorporating middle-range coverage in the training data substantially improves interpolation results. This suggests that the presence of intermediate values in the training range enhances the model's ability to generalize effectively across sparsity gaps.
> > > >
> > > > Our findings highlight an interesting phenomenon in video generation. However, quantifying the optimal sparsity level is a challenging theoretical problem and is beyond the scope of this paper, which is a compelling direction for future research.
> > > >
> > > > ## Summary
> > > >
> > > >  We hope our responses have addressed your concerns regarding VAE, pretraining, and other aspects. If you have any further questions, please feel free to reach out. Thank you for your time and thoughtful review.

---

> > > > > ### Author Response · Authors · 2024-11-24
> > > > >
> > > > > Dear Reviewer ym7u,
> > > > >
> > > > > Thank you for your constructive review on our paper.
> > > > >
> > > > > We believe that the rebuttal may address your concerns such as VAE, preatrained SOTA models, and scaling law. Given the deadline around 2 days, we would appreciate your prompt feedback to ensure any further discussion and revisions can be made timely.
> > > > >
> > > > > Thank you for your time and consideration again.
> > > > >
> > > > > Best regards,
> > > > >
> > > > > authors

---

> > > > > > ### Author Response · Authors · 2024-11-25
> > > > > >
> > > > > > Dear Reviewer ym7u,
> > > > > >
> > > > > > Thank you for your thoughtful and constructive review of our paper. We greatly appreciate the opportunity to address your concerns.
> > > > > >
> > > > > > We trust that our responses adequately clarify your questions regarding VAE, preatrained SOTA models, and scaling law. Given the approaching deadline in 40 hours, we kindly request your prompt feedback to facilitate any necessary revisions or further discussion.
> > > > > >
> > > > > > Best regards,
> > > > > > The Authors

---

> > > > ### Comment · Reviewer_ym7u · 2024-11-28
> > > >
> > > > "For example, OpenAI's Sora Technical Report [3] claims that video generation models can learn rules and act as world simulators, suggesting they might be capable of learning universal rules" Are there any other work that try to explicitly claim that video models ARE useful for learning universal rules? And furthermore, learning some spatial rules is different from claiming they are infallible world models which perfectly model physics. Even humans often fail at estimating what will happen in simple AP Physics style exam questions, which could be generalized to a video setting. Does that suggest that humans do not have world models of the world since we are incapable of learning all universal rules of the world?
> > > >
> > > > Can these video models infer the physical properties needed to feed in a real physics based simulator, aka a perfect "world model" by this paper's definition? Perhaps, perhaps not. The open question is whether the video model is able to infer these properties (which the latest set of experiments have shown to some degree), to then build an accurate world model. An model must perceive the world in order to model it, or have some perfect state representation of the world (ie. a position/velocity parameter).
> > > >
> > > > This actually led to my do some quick research and finding some related work which is not discussed such as "Learning A Physical Long-term Predictor" https://arxiv.org/abs/1703.00247  and other work on learning physical laws which actually predate "World Models". Furthermore, there is an entire field of Physical-informed Neural networks, which try to explicitly model physical parameters to combat the lack of neural networks to generalize (even when given explicit states). See the seminal paper: "Physics-informed neural networks: A deep learning framework for solving forward and inverse problems involving nonlinear partial differential equations": https://arxiv.org/abs/1711.10561 As such, this finding does not seem novel to me, especially in the relatively low-data hold-out regime. I would like to see more evidence that people are overclaiming and actually using these video models to act as world models to model data significantly outside of their distributions. I would also like to see some discussion about how these results differ from say: "Unsupervised Intuitive Physics from Past Experiences" https://arxiv.org/abs/1905.10793 . It does seem like the major limitation here is the ability to infer physical properties such as mass, etc from the visual representation. The ICLR2020 review comments from that paper sound similar to this paper https://openreview.net/forum?id=SJlOq34Kwr, although that papers come to the opposite conclusion on a toy problem.
> > > >
> > > > I am concerned that the paper does not discuss any results with pre-SORA video models on modeling physics. It does seem known in the literature  that additional annotations are needed (at least in the low data diversity setting), or helpful to help physics models of the world generalize (as noted in the PiNN literature). I would also like to say the learning the kinematics of objects, is a significantly more constrained problem that world modeling. In the peer reviewed literature, I am aware of any video models that have shown evidence of accurate (as opposed to plausible) kinematic modeling of the physical world. Note that the claims in the SORA paper not peer reviewed, and the model was only widely available for a few hours earlier this week. I would like to see some supporting evidence that this claim that video models, as they are now, and as a widely available are good physical, kinematic models for out of distribution data.
> > > >
> > > > Right now, this paper seems to be trying to debunk a suggestion of a non-peer reviewed technical report. I would like some more evidence that the belief that these models are actually good world models for complex physical kinematics is widespread in the literature to support this paper. Right now, I've only seen convincing evidence in rather constrained domains (Atari/Minecraft etc), and not in continuous world states (especially for OOD data).

---

> > > > > ### Author Response · Authors · 2024-11-29
> > > > >
> > > > > Thank you for triggering this constructive discussion.
> > > > >
> > > > > > I would like some more evidence that the belief that these models are actually good world models for complex physical kinematics is widespread in the literature;   I would like to see more evidence that people are overclaiming and actually using these video models to act as world models to model data significantly outside of their distributions
> > > > >
> > > > > We'd like to discuss two published papers that might address your concerns.
> > > > >
> > > > > Paper [1] attempts to construct a general world simulator for robotics using video generation. As stated in the introduction, "In this work, we take the first steps towards building a universal simulator (UniSim) of real-world interaction through generative modeling." This indicates an explicit aim to create a framework for **simulating physical interactions and the kinematics of robot arms** across diverse scenarios via video generation. The authors expect simulated data to be used for training other models, such as low-level control policies, because the simulations are realistic. Correctly simulating low-level control requires a deep understanding of kinematics.
> > > > >
> > > > > Similarly, Paper [2] trains a video generation model for autonomous driving, with the authors asserting that the model demonstrates strong generalization capabilities to various unseen scenarios. This claim is highlighted in both the abstract and introduction, where the potential for these models to operate in new and diverse contexts is emphasized.
> > > > >
> > > > > [1] Yang M, Du Y, Ghasemipour K, et al. Learning interactive real-world simulators[J]. ICLR 2024 outstanding paper.
> > > > > [2] Yang J, Gao S, Qiu Y, et al. Generalized predictive model for autonomous driving. CVPR 2024
> > > > >
> > > > >
> > > > > > ` learning some spatial rules is different from claiming they are infallible world models which perfectly model physics` `Does that suggest that humans do not have world models of the world since we are incapable of learning all universal rules of the world?`
> > > > >
> > > > > We believe **whether a perfect world model needs to learn universal rules depends on how one defines a world model and what is expected from it**. For humans, a world model is primarily used to predict what will happen, often relying on intuitive judgments rather than precise calculations. In many cases, such intuitive predictions are sufficient for day-to-day tasks.
> > > > > However, if we expect the world model to generate video data that is indistinguishable from real-world data—particularly for applications like robotics or autonomous driving, where the model must generate scenarios not seen in the training set—then learning universal rules and generating videos based on these rules becomes necessary.
> > > > >
> > > > > Moreover, we believe that investigating whether video generation models are able to learn physical laws itself can also be a valuable topic.  We not only concluded observations for this question, but also uncovered the underlying generalization mechanisms empirically of video generation models.  We'd appreciate it if you could re-evaluate our paper, especially taking these findings into consideration.
> > > > >
> > > > > > `other related works such as PINN`
> > > > >
> > > > > Thank you for listing these works. Since the focus of this paper is on investigating whether video generation models can learn physical laws, we have primarily discussed related works relevant to video generation. We differed some related works about "physical reasoning" to Appendix A.1. We appreciate the broader context you’ve provided, and we will incorporate the papers you mentioned into the related works section once the revision is allowed.

---

> ### Comment · Reviewer_ym7u · 2024-11-30
>
> "intuitive judgments rather than precise calculations" - one major issue with this framing is that some of the experiments such as pixel perfect reconstruction defined above would not fall under the criteria. Pixel perfect calculations seem to fall outside the realm of intuitive judgements and lean more towards precise calculations? Given this update criteria for world models, how does this experiment fit into the paper? I would not describe the situations section 5.5 as intuitive since they are so sensitive to precise calculations.
>
> "These findings suggest that relying solely on visual representations, may be inadequate for accurate physics modeling" how so, and what other modalities would you include? What do you define as a "visual representation" here. This seems like a claim unsupported by the underlying experiments here, since you are just relying on the fidelity of visual representation to learn a very precise pixel-perfect measurement, and generalizing that result to other natural laws.
>
> You also state the necessity of learning "universal rules" but I am unclear how these rules differ from the laws of kinematics. In which case, that falls into the prior body of physics based learning.

---

> > ### Author Response · Authors · 2024-12-01
> >
> > Thanks for your timely response.
> >
> > `"intuitive judgments rather than precise calculations" - one major issue with this framing is that some of the experiments such as pixel perfect reconstruction defined above would not fall under the criteria`
> >
> > Please note that "intuitive judgments" is not the primary criterion for the world model in our paper. As we emphasized earlier, "For humans, a world model ften relies on intuitive judgments. In many cases, such intuitive predictions are sufficient for day-to-day tasks. **However, if we expect the world model to generate video data that is indistinguishable from real-world data—particularly for applications like robotics or autonomous driving, where the model must generate scenarios not seen in the training set—then learning universal rules and generating videos based on these rules becomes necessary."**
> >
> > In the video generation community, a key expectation for the world model is its ability to simulate the world accurately, particularly for tasks like generating data for robotics and autonomous driving. This aligns with recent work that focuses on using learned world models to generate realistic scenarios for training [1,2]. In our paper, we set this accurate and realistic simulation as one key criterion for a world model.
> >
> > [1] Yang M, Du Y, Ghasemipour K, et al. Learning interactive real-world simulators. ICLR 2024 Outstanding Paper.
> >
> > [2] Yang J, Gao S, Qiu Y, et al. Generalized predictive model for autonomous driving. CVPR 2024.
> >
> > `what other modalities would you include? What do you define as a "visual representation" here`
> >
> > This is an open question, and the answer may depend on the specific scenario. For robotics, accurate perception often requires additional information such as depth and proprioceptive data (e.g., from sensors like IMUs). For autonomous driving, other modalities such as lidar or radar data might be essential . In our paper, we define "visual representation" as purely image or video data, without incorporating these additional sensory inputs.
> >
> > Our discussion in Sec. 5.5 is to investigate how visual representation (in contrast to exact internal states like velocity and radius) affects final results. For example, the radius inferred from image pixel values has low precision due to the discritization of image resolution, compared to the internal states represented with float numbers. The precision issue is the main cause of visual ambiguity. However, it is still possible to learn a "model world" within the low-precision visual world. We will revise the main text to make this clearer.
> >
> >
> > `how these rules differ from the laws of kinematics`
> >
> > The laws of kinematics are indeed fundamental and representative of universal rules. As we emphasized earlier, the key distinction from prior work in physics-based learning is that our paper specifically focuses on investigating whether video generation models can learn physical laws, and discuss related works relevant to video generation.
> >
> > ## Final
> >
> > We sincerely thank you once again for your valuable feedback, which has greatly helped us improve our paper. After discussing questions related to VAE, pretrained SOTA models, scaling laws, evidence of overclaiming regarding the use of video models as world models, and other points, we would like to know if there are any additional concerns or suggestions you have regarding our paper. Your insights are highly appreciated.

---

> > ### Author Response · Authors · 2024-12-02
> >
> > Dear Reviewer ym7u,
> >
> > Thank you for the effort you have put into reviewing our paper. We have greatly enjoyed our discussion with you, particularly your insightful comments.
> >
> > With the discussion session closing in 15 hours, we would like to take this final opportunity to learn from your feedback.
> >
> > We would greatly appreciate it if you could provide comments on our rebuttals and any additional questions you may have.
> >
> > Best regards,
> > The Authors

---

> > > ### Author Response · Authors · 2024-12-04
> > > **Kind Request for Reconsideration of Our Submission**
> > >
> > > Dear Reviewer ym7u,
> > >
> > > Thank you for your time and effort in reviewing our paper. We regret that we have not yet earned your support following the discussion period.
> > >
> > > As the author response window is closing, we would like to briefly revisit discussions around your concerns in the hope of gaining your support:
> > >
> > > 1. **Fine-tuning Pretrained SOTA Models**: Our additional results show that fine-tuning these models yields outcomes consistent with training from scratch, in line with existing findings.
> > >
> > > 2. **VAE Performance**: Our experiments demonstrate that the VAE preserves both kinematic and pixel-level details in simple scenarios. The inability to learn physical laws is due to the diffusion process itself, not information loss from the VAE.
> > >
> > > 3. **"Color > Size > Velocity > Shape" Hierarchy**: We tested CoGVideo’s VAE and observed the same hierarchy, indicating that this pattern is not unique to our implementation.
> > >
> > > 4. **Motivation and Relevance**: Our work addresses a key challenge in the video generation community regarding the potential of video models as world models for physical kinematics. This claim is supported by recent papers, including UniSim (ICLR 2024) and GenAD (CVPR 2024), which focus on simulating physical interactions for robotics and autonomous driving.
> > >
> > > We hope these clarifications address your concerns and kindly ask you to reconsider your evaluation of our paper.
> > >
> > > Best regards,
> > >
> > > The Authors

---

### Official Review · Reviewer_47JZ · 2024-11-04

**Soundness:** 3
**Presentation:** 3
**Contribution:** 3
**Rating:** 8
**Confidence:** 3

**Summary:**

This paper conducts a systematic investigation of video diffusion models from a physical law perspective. It explores three generalization scenarios, such as combinatorial generalization, and trains the model under various settings, such as different movement types and training data scales. The experimental results reveal several intriguing findings.

**Strengths:**

1. This paper provides a systematic investigation of video diffusion models from a physical law perspective, supported by extensive experiments and solid analysis.

2. This paper shows several intriguing findings, improving the understanding of physical laws in video diffusion models.

3. The paper is well-organized and clearly written.

**Weaknesses:**

1. This paper investigates physical laws by training video diffusion models on specifically designed data.  How would the findings apply to open-source pretrained models, such as Stable Video Diffusion [1]?

[1] Stable Video Diffusion: Scaling Latent Video Diffusion Models to Large Datasets

2. Based on the analysis in this paper, are there guidelines for improving the ability of video diffusion models to learn physical laws?

**Questions:**

Please refer to the above weaknesses.

---

> ### Author Response · Authors · 2024-11-22
>
> Thank you for your time and thoughtful review. For your questions, please see the response below.
>
> `Q1: This paper investigates physical laws by training video diffusion models on specifically designed data. How would the findings apply to open-source pretrained models, such as Stable Video Diffusion?`
>
> Please refer to the global response for a detailed discussion on this question.
>
> `Q2: Based on the analysis in this paper, are there guidelines for improving the ability of video diffusion models to learn physical laws?`
>
> 1. **Improve combinatorial diversity**: Our findings demonstrate that video generation models exhibit combinatorial generalization, and the model’s adherence to physical laws improves significantly as the combinatorial diversity of the training data increases. Thus, designing training datasets with greater combinatorial diversity over physical interactions is a key guideline.
>
> 2. **Develop kinematic-aware video representations**: Our analysis reveals that video generation models primarily operate through a "data retrieval" process, following a retrieval priority order of color > size > velocity > shape. This behavior appears to stem from the structure of the VAE latent space, which is primarily pixel-driven and lacks explicit representations of objects or kinematics (for detailed analysis see appendix A6.2). By incorporating geometry-aware or kinematic-aware latent representations, models could better encode spatial and temporal relationships .
>
> 3. **Integrate predictive priors**: Since our study indicates that current models largely perform memorization without inherent physical reasoning, a promising approach is to incorporate predictive priors. For example, using LLMs or physics simulators to predict latent variables of physical dynamics over time could allow diffusion models to focus solely on rendering.
>
> We hope this response addresses your questions. If you have any further inquiries, please don't hesitate to reach out.

---

> > ### Comment · Reviewer_47JZ · 2024-11-26
> >
> > Thanks to the authors for their rebuttal. My questions have been well answered, and I will maintain my rating.

---

> > > ### Author Response · Authors · 2024-12-02
> > >
> > > Dear Reviewer 47JZ,
> > >
> > > Thank you for the effort you have put into reviewing our paper and for your support.
> > >
> > > With the discussion session closing in 15 hours, we would like to take this final opportunity to learn from your feedback.
> > >
> > > Best regards,
> > > The Authors

---

> ### Author Response · Authors · 2024-11-26
>
> Thank you for taking the time to review our paper and for your support for its acceptance. We hope our rebuttal enhances your confidence in our paper. If so, we wonder if this can be reflected in an increased confidence score. We would be happy to provide any additional materials if needed.

---

### Author Response · Authors · 2024-11-22
**Global Response**

We sincerely thank the reviewers for their constructive feedback and thoughtful questions. We have provided a revised PDF with newly added sections highlighted in blue, primarily in the appendix. Below, we address the common concerns:

`Q1: Why train a video gen model from scratch instead of finetuning pretrained SOTA models?`

Several reviewers questioned the rationale behind training a model from scratch. Here, we provide a detailed explanation:

The primary goal of this paper is to answer the fundamental question: “Can a video generation model learn physical laws?” Existing pretrained models are built on large-scale datasets sourced from publicly available internet content. However, the extent to which these datasets contain videos about physical laws or structured physical interactions remains unknown. This lack of transparency leads to critical concern - **contamination of evaluation data**: There is a risk that pretrained models may have been exposed to data similar to our evaluation experiments during training. This potential overlap undermines the validity of assessing their ability to learn physical reasoning.

To address these concerns, **we chose to train a model from scratch, which provides complete control over the training data. This approach allows us to precisely ensure that the results stem from the model’s actual learning process rather than potential uncontrollable pretraining data.**

This methodology is supported by recent research into the mechanisms of language models [1, 2], where training relatively small models from scratch has proven effective in isolating and analyzing reasoning processes more rigorously.

`Q2: How would the findings in the paper apply to finetuning open-source pretrained models, such as Stable Video Diffusion (SVD)?`

To evaluate whether the inability to learn physical laws is specific to our setup or if it also applies to open-source pretrained models, we conducted experiments using the Stable Video Diffusion (SVD) model on a representative uniform motion task. Specifically, we fine-tuned SVD for 300k steps with 3M uniform motion videos, mirroring the training setup of DiT-B in our experiments. Below are the key observations and conclusions:

1. **VAE Reconstruction Error**: The reconstruction error of SVD’s VAE is very small, indicating that its VAE component can effectively preserve kinematic information.
2. **Diffusion Model Performance**: Despite pretraining, the diffusion component of SVD performs worse than DiT-B in both in-distribution (ID) and out-of-distribution (OOD) predictions. We hypothesize this is due to the significant gap between the pretraining dataset (internet videos) and the simple kinematic test scenarios used in this evaluation.
3. **Robustness of OOD Error Trend**: The OOD error for SVD is an order of magnitude larger than its ID error, similar to the trend observed with DiT-B. This reinforces the conclusion in our paper that the inability of video generation models to generalize to OOD scenarios is not resolved by pretraining and is robust across different architectures, including open-source models like SVD.

| Model                     | ID Error | OOD Error |
|---------------------------|----------|-----------|
| GT                        | 0.0099   | 0.0104    |
| DiT-B (from scratch, ours)| 0.0138   | 0.3583    |
| SVD-VAE-Recon             | 0.0103   | 0.0107    |
| SVD-Finetune              | 0.0505   | 0.9081    |

These results confirm that our findings about the challenges of learning physical laws in video generation models are not limited to a specific architecture but are applicable to open-source pretrained models like SVD.


[1] Ye, Tian et al. “Physics of Language Models: Part 2.1, Grade-School Math and the Hidden Reasoning Process.” ArXiv abs/2407.20311 (2024).

[2] Allen-Zhu, Zeyuan and Yuanzhi Li. “Physics of Language Models: Part 3.1, Knowledge Storage and Extraction.” ICML 2024.

---

### Meta-Review · Area_Chair_s676 · 2024-12-20

**Metareview:**

This paper studies generalisation properties of a DiT-style video diffusion model trained on synthetic 2D simple physics simulation videos using the PHYRE simulator.

The reviewer reception of this paper was mixed: reviewers positively highlight the quality of writing and the structured analysis of in-distribution vs. out-of-distribution generalisation on this particular dataset.

The concerns brought forward by the rejection-leaning reviewers, however, are compelling and the AC agrees with them: the paper makes strong claims about scaling properties of this class of models but test them in only very simplistic settings, which do not fully support the broad claims made in the motivation of the paper. For a future resubmission, it would be valuable to expand the analysis to more complex scenarios (incl. using pre-trained models to study their adaptation capabilities) and/or make more specific claims as to what the analysis addresses.

**Additional Comments On Reviewer Discussion:**

No reviewer was willing to champion the paper for acceptance during the discussion period.

---

### Decision · Program_Chairs · 2025-01-22

Reject